

# Late Pleistocene glacial chronologies and paleoclimate in the
# northern Rocky Mountains
**Brendon J. Quirk[1*], Elizabeth Huss[2], Benjamin J.C. Laabs[3], Eric Leonard[4], Joseph**
**Licciardi[5], Mitchell A. Plummer[6], Marc W. Caffee[7]**
*[1] Department of Geology and Geophysics, University of Utah, Salt Lake City, Utah 84112, USA*
*[2]Department of Geosciences, State University of New York at Geneseo, Geneseo, NY 14454, USA*
*[3]Department of Geosciences, North Dakota State University, Fargo, North Dakota 68102, USA*
*[4]Department of Geology, Colorado College, Colorado Springs, Colorado 80903, USA*
*[5]Department of Earth Sciences, University of New Hampshire, Durham, New Hampshire 03824,*
*USA*
*[6]Idaho National Engineering and Environmental Laboratory, Idaho Falls, Idaho 83415, USA*
*[7]Department of Physics, PRIME Lab, Purdue University, West Lafayette, Indiana 47905, USA*
*[*]Corresponding author email address: bjquirk@purdue.edu*
**ABSTRACT**
The geologic record of mountain glaciations is a robust indicator of terrestrial paleoclimate
change. During the last glaciation, mountain ranges across the western U.S. hosted glaciers while
the Cordilleran and Laurentide ice sheets flowed to the west and east of the continental divide,
respectively. Records detailing the chronologies and paleoclimate significance of these ice
advances have been developed for many sites across North America. However, relatively few
glacial records have been developed for mountain glaciers in the northern Rocky Mountains near
ice sheet margins. Here, we report cosmogenic beryllium-10 surface exposure ages and
numerical glacier modeling results showing that mountain glaciers in the northern Rockies



abandoned terminal moraines after the end of the Last Glacial Maximum around 17-18 ka and
could have been sustained by -10 to -8.5°C temperature depressions relative to modern assuming
similar or drier than modern precipitation. Additionally, we present a deglacial chronology from
the northern Rocky Mountains that indicates while there is considerable variability in initial
moraine abandonment ages across the Rocky Mountains, the pace of subsequent ice retreat
through the Lateglacial exhibits some regional coherence. Our results provide insight on
potential regional mechanisms driving the initiation of and sustained deglaciation in the western
U.S. including rising atmospheric $CO_2$ and ice sheet collapse.
**INTRODUCTION**

Mountain glaciers are widely recognized as robust indicators of modern climate change

(Oerlemans, 2005; Vaughan et al., 2013; Mark and Fernández, 2017). Investigations of past glacier
fluctuations preserved in the geologic record can therefore reveal valuable information regarding
past climate oscillations and variability (e.g. Gilbert, 1890; Blackwelder, 1931; McCoy et al.,
1985; Marcott et al., 2019). In the Rocky Mountain region of the western U.S. records of mountain
glaciation have been used extensively to reconstruct the regional pattern of Pleistocene glaciation
in space and time (e.g., Porter et al., 1983; Leonard, 1989; Licciardi et al., 2004; Laabs et al., 2009;
Quirk et al., 2020), but few studies have focused on northern ranges along the former southern
margins of the Laurentide and Cordilleran ice sheets. While surficial geologic records of
Pleistocene mountain glaciation in the northern Rocky Mountains of western Montana have been
available for decades (Alden, 1932; Carrara, 1987), these records have seldom been used to infer
Pleistocene climate (e.g., Murray and Locke, 1989). Many ranges were occupied by coalesced
valley glaciers and ice caps with high-altitude ice divides, which are especially difficult to
reconstruct based solely on mapped glacial deposits and landforms. Additionally, in much of



northwestern Montana, mountain glaciers likely coalesced with the southern edges of the
Laurentide and Cordilleran ice sheets, which also complicates reconstructions of paleo-glaciers,
and limits the usefulness of traditional methods for inferring past climate from glacier equilibrium-
line altitudes or mass-balance gradients.
However, discrete Pleistocene mountain glaciers occupied some ranges of western
Montana, as evidenced by a well-preserved record of deposits and landforms delimiting their
maximum extent during the last glaciation. Such records are found in the northern Absaroka Range
in southwestern Montana and the eastern Lewis Range in northwestern Montana (Figure 1), where
glaciers incised deep valleys and in some areas constructed broad terminal moraine complexes
along mountain fronts. These records present an opportunity to reconstruct mountain glacier
extents and develop cosmogenic chronologies of the last glaciation. These spatiotemporally
constrained paleo-glaciers can then, in turn, be used to infer paleoclimate conditions in the northern
Rocky Mountains during the last glaciation.
Here we present new surficial mapping of latero-terminal moraines of the last Pleistocene
glaciation in the Cut Bank and Lake Creek valleys in the eastern Lewis Range and cosmogenic
[10]Be surface exposure ages of a terminal moraine complex in Cut Bank valley. For the northern
Absaroka Range, we present new exposure ages for latero-terminal moraines in South Fork Deep
Creek and Cascade Creek valleys as well as glacially scoured bedrock ages from Pine Creek to
track ice retreat from a previously dated terminal moraine to a cirque floor. We use the
spatiotemporal glacial histories from the Lewis and Absaroka ranges to inform numerical
modeling of paleo-glacier shapes, thicknesses and paleoclimate conditions (i.e., precipitation and
temperature) for mapped and dated glacial stadials. We then compare the glacial chronologies and
glacier-climate modeling results developed for the Lewis and northern Absaroka Ranges to those



from other western North America mountain ranges and examine how these glacial histories can
inform our understanding of regional patterns of glaciation and climate change.

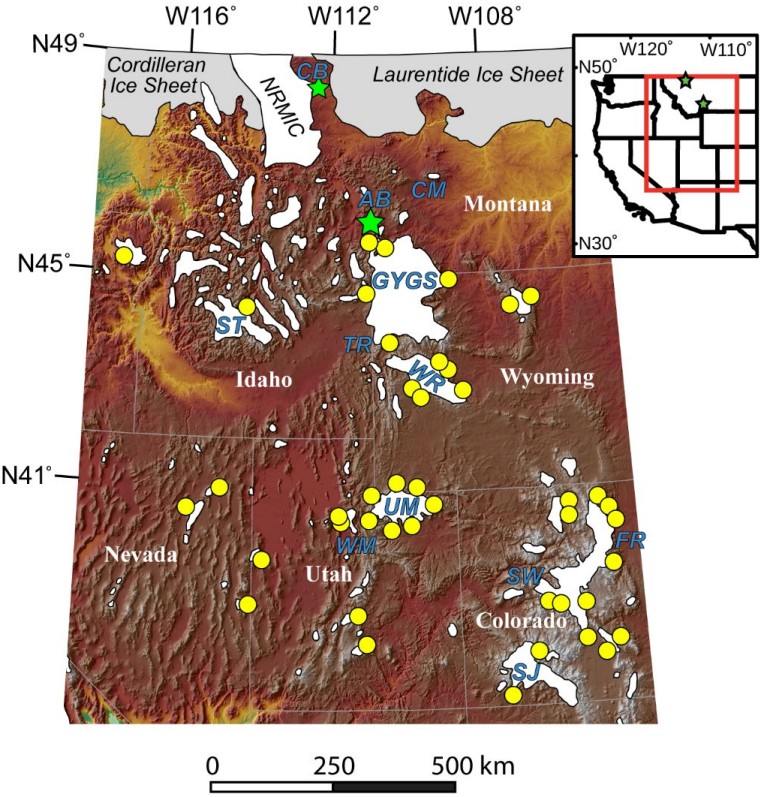


*Figure 1. Pleistocene ice extents in the northern U.S. Rocky Mountains (after Pierce et al., 1983;*
*Pierce, 2003) with the locations of our two field sites, Cut Bank Creek (CB) in the Lewis Range*
*and Pine Creek, South Fork Deep Creek, and Cascade Creek in the northern Absaroka Range*
*(AB) indicated by green stars. Locations of previously established age control are indicated by*
*yellow circles including the Greater Yellowstone glacial system (GYGS), Crazy Mountains (CM),*
*Wind River (WR), Sawtooth (ST), Wasatch (WM), Uinta (UM), Front Range (FR), Sawatch (SW),*
*and San Juan (SJ) ranges. General outlines of the Cordilleran and Laurentide ice sheets as well*
*the northern Rocky Mountain ice cap (NRMIC) are also shown. (Inset) Map of Western North*



America with state outlines. Green stars indicate our study areas and the red box shows the
approximate coverage of the main illustration.

**Site Description**

The Lewis Range hosted numerous glaciers and, in some areas, coalesced forming the

northern Rocky Mountain ice cap (Figure 1). In this study, we focus on the Cut Bank Creek
glacier which flowed east from its headwaters at 2.6 km asl and terminated on the piedmont just
above 1.4 km asl at its maximum extent. The Cut Bank glacier did not coalesce with either the
northern Rocky Mountain ice cap to the west and north or the Laurentide ice sheet to the east
during Pinedale times and flowed as a discrete mountain glacier. The glacier was over 25 km
long at its maximum extent and in many areas was over 200 meters thick with maximum ice
thickness in excess of 300 meters.

The Absaroka Range, located to the north of the Greater Yellowstone glacial system

(Figure 1) also hosted several glaciers during Pinedale times including the Pine Creek, South
Fork Deep Creek, and Cascade Creek glaciers. The three glaciers flowed from southeast to the
northwest just to the range front where they built terminal and lateral moraine complexes. All
three canyons have headwaters at or above 3 km asl and generally flowed down to elevations of
around 1.6-1.7 km asl. The Pine Creek Pinedale glacier was the longest of the three at over 13
km at it's maximum extent. The Cascade and South Fork Deep Creek glaciers were around 6 and
7 km long at their maximum Pinedale extents, respectively. Ice thicknesses were thinner in the
Absaroka Range glaciers as compared to the Cut Bank glacier, with many areas hosting 100-200
meter thick ice and maximum thicknesses in Pine Creek of 250-300 meters.



**Previous Studies**

Reconstructions of Pleistocene glaciers in the northern Rocky Mountains of western Montana are limited (Pierce, 2003), and relatively little work has been done inferring past climate in the region from paleoglacier characteristics. Most previous work has focused either on the Greater Yellowstone area of southern Montana or on the Glacier National Park area of northern Montana – also the foci of the current study. In these and other areas of western Montana past workers have identified deposits and landforms from the penultimate and most recent glaciations, generally termed Bull Lake and Pinedale glaciations, following the terminology developed by Blackwelder (1915) for the Wind River Range of Wyoming (Fig. 1). Based on chronologies of glacial deposits throughout the Middle and Southern Rocky Mountains, these last two Pleistocene glaciations are thought to correspond broadly with intervals of global ice volume increase during marine isotope stages (MIS) 2 and 6, respectively (Licciardi and Pierce, 2008; Licciardi and Pierce, 2018; Quirk et al., 2018; Dahms et al. 2018; Schweinsberg et al., 2020; Laabs et al., 2020). Chronological work utilizing cosmogenic nuclide surface-exposure dating in the Yellowstone/Grand Teton National Parks area of southwestern Montana and adjacent northwestern Wyoming (Licciardi et al., 2001; Licciardi and Pierce, 2008, 2018; Pierce et al., 2018) has allowed subdivision of Pinedale-age deposits as is discussed below.

Deposits of Pleistocene mountain glaciers in the eastern Lewis Range of western Montana were mapped and described as early as 1906 by Calhoun and then later by Alden (1932), Carrara (1989), and Fullerton et al. (2004). Calhoun (1906) described the broad hummocky terminal and recessional moraines deposited on the plains to the east of Cut Bank Creek headwaters investigated in this study as well as several recessional moraine ridges deposited up valley. Fullerton et al. (2004) identified multiple Pinedale tills, two ages of Bull Lake till, and a possible pre-Bull Lake

till in moraine deposits at Cut Bank Creek and elsewhere along the eastern front of the Lewis
Range. No numerical ages are available for these deposits, although a radiocarbon age on a wood
fragment, underlying two latest Pleistocene tephra layers in lake sediment at Marias Pass, provides
a minimum age of 12,194±145 [14]C yr (Carrara, 1995) or ~14,245 cal yr (Fullerton et al., 2004) for
complete recession of at least one east-side outlet glacier of the Northern Montana Ice Cap in the
Glacier National Park region.

Pleistocene glacial deposits north of Yellowstone National Park and near the northern

Absaroka Range were first described and mapped by Weed (1893) and then later by Pierce (1973;
1979 and references therein). Licciardi and Pierce (2018) identified three distinct phases of
glaciation in the Greater Yellowstone region during the last glacial including the early (22-18 ka),
middle (18-16 ka), and late (16-13 ka) Pinedale. While the early Pinedale phase in the Yellowstone
area occurred mainly during the interval of the global Last Glacial Maximum (26.5-19.0 ka; Clark
et al., 2009); the middle and late Pinedale phases clearly postdated the global LGM, although they
appear to have predated the Younger Dryas interval. Terminal and recessional moraines at the
southwestern front of the northern Absaroka Range and in the neighboring Paradise Valley to the
south of cosmogenic [10]Be exposure ages were originally reported by Licciardi et al. (2001) and
combined with additional data by Licciardi and Pierce (2008; 2018). The terminal moraine in Pine
Creek valley of the northern Absaroka Range has a mean cosmogenic [10]Be exposure age of 18.2
± 0.5 ka (± 1 standard error of the mean) as recalculated using methods described in the text. In
Paradise Valley, moraines delimiting the terminus of the northern outlet of the Yellowstone glacial
system have mean 10Be exposure ages of 17.9 ± 0.4 ka for the Eightmile terminal moraine and
17.1 ± 0.6 ka for the Chico recessional moraine. Together, these exposure ages indicate that
mountain glaciers in this sector of the Greater Yellowstone glacial system began retreating from



their terminal moraines during the middle Pinedale and, critically, after the end of the global Last
Glacial Maximum.

Less attention has been paid by previous workers to use of paleoglaciological methods to

reconstruct late Pleistocene climate in western Montana than to reconstruction of the extent and
chronology of past glaciation. Locke (1990) examined modern and reconstructed late Pleistocene
glacier equilibrium lines throughout western Montana, concluding that an assumed late Pleistocene
temperature depression of 10°C would have been associated with decreased precipitation relative
to the present. Based on mapping of glacial deposits and landforms in the Crazy Mountains of
southwestern Montana (Figure 1), Murray and Locke (1989) reconstructed the geometry and ice
flux of a valley glacier in Big Timber Canyon. They interpret the reconstructed ice-surface gradient
and ice flux as indicators of a cold and dry regional climate during the last glaciation. Hostetler &
Clark (1997) used a combination of climate-model output and glacier equilibrium-line modeling
and concluded that during the LGM in the Yellowstone region summer temperatures were 10-
15°C colder than present with winter precipitation approximately equal to present, while in
northern Montana winter temperature depression was even greater but precipitation was reduced
compared to modern.

Nowhere in the U.S. northern Rocky Mountains have more recent

paleoglaciological methods, particularly distributed energy/mass-balance models or degree-day
mass-balance models, been applied to reconstructed late Pleistocene glaciers, as they have been
successfully applied in the Middle Rocky Mountains (Laabs et al., 2006; Refsnider et al., 2008;
Birkel et al., 2012; Quirk et al., 2018, 2020) and Southern Rocky Mountains (Ward et al., 2009;
Brugger, 2010; Brugger et al. 2018, 2019; Dühnforth and Anderson, 2011; Leonard et al., 2014,
2017a; Schweinsberg et al., 2016;). In this study we apply a modified version of the Plummer and



Phillips (2003) distributed energy/mass-balance model to reconstructed glaciers in the Absaroka
and Lewis ranges to help elucidate climate conditions in the northern Rockies during the last
glaciation.

.

**METHODS**
**Moraine Mapping**

Although terminal moraines of east-flowing glaciers in the Lewis Range are known from

previous studies, they were remapped here to aid with reconstructing maximum ice extent in the
Cut Bank Creek and Lake Creek valleys (Figure 2a). Moraines in both valleys were examined in
aerial imagery available in Google Earth and using 1:24,000-scale topographic maps. The portion
of the terminal moraine north of Cut Bank Creek was mapped in the field. Moraines were identified
as broad (0.5-1 km wide), looping plateaus with hummocky topography (Figure 3) on the piedmont
east of the Lewis Range and featured abundant erratic boulders at their crests.

Surficial mapping of glacial deposits within our area of interest in the Absaroka Range had

been previously completed by Pierce (1979 and references therein). Mapping in the Pine Creek
area was subsequently updated by Licciardi and Pierce (2008). In the field, we checked and
confirmed, without modification, the moraine mapping from these previous studies.






*Figure 2. (Top) Cut Bank Creek study area located in the Lewis Range of northern Montana.*
*(Bottom) northern Absaroka Range study area including South Fork Deep Creek, Pine Creek, and*
*Cascade Creek drainages. Pinedale maximum ice extents are outlined in black (dashed where*
*inferred). Recessional position at Cut Bank outlined in light-grey. Moraine deposits are shown in*
*yellow with cosmogenic $^{10}$Be boulder sample locations indicated by the red circles with exposure*
*ages and analytical uncertainty (shown in ka) and sample codes in blue text. Exposure ages*
*interpreted as outliers are shown in grey and italicized.*






*Figure 3. (A) Characteristic hummocky morphology of Cut Bank Creek terminal moraine. (B)*

*Boulder CB-03 targeted for cosmogenic exposure dating on the Cut Bank terminal moraine. (C)*

*Photograph taken facing north-northeast looking across the threshold of Pine Creek Lake and*

*towards bedrock sampled for cosmogenic exposure dating. (D) Location of bedrock sample PC11-*





*11. (E) Lateral sector of the Cascade Canyon Pinedale terminal moraine. (F) South Fork Deep*

*Creek lateral moraine sample DC12-01.*

**Cosmogenic [10]Be Exposure Dating**

Following moraine mapping and field verification, we selected moraines and erratic boulders atop moraine crests for *in-situ* cosmogenic [10]Be exposure dating to determine landform ages. at Cut Bank Creek, South Fork Deep Creek, and Cascade Creek canyons. Boulders atop a recessional moraine identified just beyond the mouth of Cut Bank Canyon were also sampled to limit the time when moraine building at the mountain front ceased and ice retreat commenced.

On moraine crests, we searched for large (>0.5 m tall), quartz-bearing boulders with broad horizontal surfaces. When possible, we selected boulders and bedrock surfaces with clear glacial polish and/or striations. In the northern Absaroka Range, most sampled moraine-boulders consisted of Jewel Quartzite (Archean rocks of the Wyoming Province; Zientek et al., 2005), which generally contains >90% quartz and some accessory minerals. In the Cut Bank Creek valley, sampled moraine boulders consisted of silica-cemented quartz arenite derived from the Appekunny Formation (subdivided from the Proterozoic Belt Supergroup), which is widely exposed along bedrock divides in the Lewis Range (Whipple et al., 1984). By selecting only samples with clear glacial polish and/or striations we determined that sample surface erosion was insignificant, and we therefore used an erosion rate of 0 cm a$^{-1}$ in exposure age calculations. Samples were collected using a hammer and chisel to depths ranging from 1 – 5 cm, with an average depth of 3 cm. The number of samples collected from each landform varied based on the availability of suitable targets. Topographic shielding data were collected in the field with a clinometer. Target surfaces were selected so as to minimize the effect of internal shielding and cosmic ray scattering from nearby boulders.





At Pine Creek in the northern Absaroka Range, where cosmogenic [10]Be exposure ages of
latero-terminal moraines were already available (Licciardi and Pierce, 2008), glacially scoured
bedrock and erratic boulders were sampled along the path of ice retreat. Here, we assume that
bedrock surfaces became progressively exposed through time as ice retreated up valley from the
terminal moraine and, therefore, exposure ages would limit the pace and timing of ice retreat (cf.
Guido et al., 2007). Jewel Quartzite, described above, bedrock and erratic boulders were sampled
along the length of the transect and were collected following the same procedure described above.
All samples were prepared at SUNY Geneseo for in-situ cosmogenic [10]Be measurement
following methods in Laabs et al. (2013). Samples were crushed, milled, and sieved to a target
grain size of 250-500 μm. Quartz grains were isolated using a rare earth hand magnet, Franz
magnetic separator, density separation, and dilute acid treatment. The quartz purification process
was accomplished by repeated etching in dilute hydrofluoric and nitric acids (Kohl and Nishiizumi,
1992). Prior to dissolution in concentrated hydrofluoric acid, the purified quartz fraction of each
sample was spiked with a commercially made [9]Be carrier solution purchased from SPEX CertiPrep
with a certified Be concentration of 1 mg/mL. Procedural blanks were prepared using equal carrier
mass as was added to samples. The beryllium fraction of each sample was chemically isolated and
loaded into targets for [10]Be/[9]Be measurement by accelerator mass spectrometry (AMS) at the
Purdue University Rare Isotope Measurement Laboratory (Sharma et al., 2000; Muzikar et al.,
2003). All [10]Be/[9]Be values were normalized to the AMS beryllium standard 07KNSTD
(Nishiizumi et al., 2007).
We calculated cosmogenic [10]Be exposure ages using the Balco et al. (2008) online
exposure age calculator, version 3.0 (http://hess.ess.washington.edu/math/). This calculator and
version were selected because they implement the Lifton-Sato-Dunai nuclide dependent (LSDn;





Lifton et al., 2014) scaling model and production rates based on user-defined calibration data from
independently dated locations. Production rates were computed using *in situ* [10]Be data from the
independently dated surface at the Promontory Point production-rate calibration site reported by
Lifton et al. (2015), which features well-preserved and continuously exposed surfaces following
the Bonneville Flood at 18,350 ± 300 cal. yr BP. We chose this calibration site because of its
proximity in space and time to the study area, following other recent reports of Pleistocene moraine
chronologies in the Rocky Mountains (Licciardi and Pierce, 2018; Schweinsberg et al., 2020;
Laabs et al., 2020). Moraine ages and associated uncertainties are reported as the arithmetic mean
of individual boulder exposure ages and the standard error of the mean, respectively (as in Putnam
et al., 2010; Quirk et al., 2020).
**Glacier Modeling**
The coupled energy/mass-balance and ice-flow models used in this study were originally
developed by Plummer and Phillips (2003) and have been successfully used to estimate
paleoclimate conditions for extinct glaciers in a variety of geologic settings (Quirk et al., 2020;
Rowan et al., 2014; Leonard et al., 2014; Harrison et al., 2014; Laabs et al., 2006). Additionally,
several studies have verified the model's ability to successfully predict snow accumulation (Laabs
et al., 2006; Leonard et al., 2014) and melt (Quirk et al., 2020), as well as small glacier extents
(Plummer, 2002) for modern conditions in the western U.S.
Our modeling approach is to match simulated glacier extents produced under prescribed
climate perturbations relative to modern (e.g., temperature depression and precipitation change) to
field evidence such as terminal and lateral moraines. In this study, we match modeled glacier
shapes and thicknesses to the well-defined Pinedale maximum ice extents at Cut Bank, Pine Creek
and South Fork Deep Creek. In order to test the validity of the ice flow parameters used for the





Cut Bank Creek glacier detailed below, we reconstructed the undated Lake Creek glacier
immediately to south of Cut Bank at its maximum mapped extent using the same parameters. We
reconstructed the Cut Bank glacier using a model spatial resolution of 180 m while we used a
resolution of 30 m for the Pine Creek and South Fork Deep Creek glaciers, which were modeled
in the same domain (herein the northern Absaroka domain). We did not include Cascade Creek as
a target for glacier reconstructions because mapping of the glacier's exact terminal position
remains unresolved.

The energy/mass-balance model calculates snow accumulation and ablation at every cell

within the model domain for the time interval of interest, typically one to several years. Annual
mass balance depends mostly on precipitation and temperature, which are the principal inputs to
the model. In this study, we use a similar approach to the one used by Leonard et al. (2017a)
whereby we describe the monthly spatial distribution of temperature and precipitation at every cell
across the model domain with linear regressions of elevation and PRISM (Parameter-elevation
Regression on Independent Slopes Model http://www.prism.oregonstate.edu/; Daly et al., 2008)
monthly mean climatological models. Secondary climate parameters include estimates of average
monthly relative humidity, cloudiness, and wind speed, and are taken directly or derived from a
combination of RAWS and NOAA COOP Station historical weather station data. We calculated
average monthly cloudiness for the Cut Bank and Lake Creek Canyon domains by determining the
fraction of days per month with precipitation (i.e. 0.5 cloudiness = 15 days of precipitation / 30
days total). For the Pine and South Fork Deep Creek domain, cloudiness was estimated using the
ERA-Interim    3[rd]    generation    (1979-2015)    reanalysis    (http://cci-
reanalyzer.org/reanalysis/monthly_tseries/). Wind speed (Ws) was scaled for elevation from





weather station data using a given weather station's reference elevation (Elevation $_{REF}$) using the
equation:

(1) $W_s = W_{SREF} + (Elevation - Elevation_{REF}) * k$

where $W_{SREF}$ is wind speed at the reference elevation and k is a wind scaling factor. Here, k is
taken as 0.001, resulting in an additional 1 m s$^{-1}$ average wind speed per 1000 m elevation. Average
monthly cloudiness is held constant at every cell and elevation within the model domain. To
simulate paleo-glacier extents, we varied precipitation and temperature, the two dominant climate
input parameters, using multiplicative and additive variations from modern, respectively. Thus, a
precipitation factor change of 1 is equal to modern precipitation and a temperature of depression
of 0 °C is modern temperature.

The primary output from the energy/mass-balance model is a mass-balance grid for model

domain. The mass-balance grid is input to the ice-flow model along with a digital elevation model
of the drainage basins. The ice-flow model designed by Plummer and Phillips (2003) used here is
similar to the finite-element ice sheet model described by Fastook and Chapman (1989) and
follows the commonly used shallow-ice approximation. Snow and ice mass is gained in the
accumulation zone and flows along the ice-surface gradient via deformation and sliding into the
ablation zone. We run glacier simulations to steady-state where the simulated terminus stabilizes
at a mapped moraine position. We define steady-state condition in our model runs as when the
integrated surface balance errors are less than 5%, and typically ≈ 0%, as described by Plummer
and Phillips (2003). The time-dependent ice-flow model is an alternating direction-implicit, space
-entered, finite-difference form of the continuity equation for 2-D flow:

(2) $\partial h/\partial t = b_n - \partial q_x/\partial x - \partial q_y/\partial y$





where h = ice-surface elevation, $b_n$ = net annual mass balance, q = ice discharge per unit width,
and x and y are orthogonal directions of ice flow in the horizontal plane. The ice flux between
neighboring cells is determined by the thickness and depth-integrated flow velocity, U, which is
the sum of ice flow via deformation and sliding:
(3) $U = u_d + u_s = (1-f) H\ 2/5\ (\tau A)^m + f\ (\tau B)^n$
Here A is the deformation flow coefficient, B is the sliding flow coefficient, H is ice
thickness, f is a velocity scaling parameter, and $\tau$ is basal shear stress. The exponents m and n are
taken to be 3 and 2, respectively, as described by Fastook and Chapman (1989). We tuned ice flow
parameters A, B, and f to match simulated glacier shapes and ice thicknesses to the observational
record. Ice flow parameter values that simulated observed ice thicknesses well included A values
for the Cut Bank and northern Absaroka domains of 8.0 E-5 $a^{-1}$ $kPa^{-3}$ and 1.0 E-7 $a^{-1}$ $kPa^{-3}$, and B
values of 0.0015 m $a^{-1}$ $kPa^{-2}$, and f values of 0 and 0.5, respectively. The ice-flow parameters used
in northern Absaroka domain agree well with the published range of values used in previous glacier
flow models (Oerlemans, 1989; Plummer and Phillips, 2003; Laabs et al., 2006; Quirk et al., 2018).
The Cut Bank glacier required a greater value of the deformation flow coefficient compared
to the steeper valley glaciers in the northern Absaroka Range. Although it is likely that the Cut
Bank Creek glacier was sliding at its base, we did not account for the contribution of sliding to
flow because it was likely far less than the contribution to flow by deformation as indicated by the
great ice thicknesses and low surface slopes. As described previously, we also simulated the Lake
Creek glacier immediately south of Cut Bank using identical ice-flow parameters to test the
validity of the chosen values. Through experimentation, we tuned the ice-flow parameters to
produce simulated steady-state glaciers that matched the mapped paleo-glacier thickness and shape





in both valleys and thus parameterized the effects the piedmont lobe and glacier shape had on the
Cut Bank glacier. (Supplemental Figure 1).
**RESULTS**
**Moraine Mapping**
The suite of moraines deposited at the mountain front in Cut Bank Creek valley features
three broad, looping plateaus with hummocky topography separated by incised meltwater channels
and outwash. The suite includes a multi-crested terminal moraine deposited farthest beyond the
mountain front and a recessional moraine deposited near the mouth of Cut Bank Canyon (Figure
2). The ice-distal sector of the terminal moraine has the highest internal relief (up to 30 m) along
the portion of the moraine south of Cut Bank Creek, with numerous closed depressions, some of
which are filled with shallow lakes. The distal slope of the moraine grades to a broad, gently
sloping outwash plain known locally as Starr School Flat, featuring low-relief (<3 m) depressions
and abandoned braided channels. The ice-proximal sector of the terminal moraine is narrower with
less internal relief (less than 15 m) and fewer closed depressions. The proximal slope of this sector
of the moraine appears to be partially buried by outwash where it is bisected by Cut Bank Creek.
The recessional moraine is best preserved north of Cut Bank Creek and features low-relief
hummocky topography (less than 5 m internally). In Lake Creek valley, only a single, looping
terminal-moraine ridge is preserved at the mountain front, forming a broad area of hummocky
topography with greater internal relief (up to 60 m along portions north Lake Creek).
The moraines delimit the size and shape of the piedmont lobes formed by glaciers in the
two valleys. In Cut Bank Creek valley, the piedmont lobe had a maximum diameter of 6.8 km
while occupying the distal sector of the moraine. While occupying the ice-proximal sector, the
piedmont lobe was reduced in diameter to approximately 4.4 km and likely became thinner or



formed a more gradual slope near the terminus as evidenced by the lower relief along the moraine.
The piedmont glacier width was further diminished upon retreat to the recessional moraine to
approximately 1.3 km, only slightly wider than the mouth of Cut Bank Canyon. In Lake Creek
valley, the piedmont lobe formed an irregular shape, likely due to partial confinement of the
northern side of the lobe by the right-lateral moraine in the neighboring Cut Bank Creek valley.
The piedmont lobe had a maximum width of about 2.5 km when the terminal moraine was
occupied. Upvalley of the terminal moraines in Cut Bank Creek and Lake Creek valleys, lateral
moraines and other glacial features mapped by Carrara (1989) were used to delimit ice thickness
and areal extent.
**Cosmogenic $^{10}$Be Exposure Ages**

Here we present 29 cosmogenic $^{10}$Be exposure ages collected from glaciated catchments

in the Lewis and northern Absaroka Ranges of Montana (Figure 2; Figure 4). In the Lewis Range,
nine exposure ages are from the ice-distal sector of the terminal moraine in Cut Bank Creek valley
and four are from a recessional moraine up valley. In the northern Absaroka Range, two samples
are from the Cascade Creek lateral moraine, three are from the South Fork Deep Creek lateral
moraine, and eleven are Pine Creek Canyon bedrock and erratic samples. The $^{10}$Be/$^{9}$Be ratios in
procedural blanks ranged from $6.00 \times 10^{-15}$ to $4.90 \times 10^{-14}$. Sample $^{10}$Be/$^{9}$Be ratios ranged from
$3.18 \times 10^{-13}$ to $1.37 \times 10^{-12}$ (Supplemental Table 1). The range of AMS measurement uncertainties
(one sigma) for most samples was approximately 1.5 – 3.5%. Both moraine-boulder samples from
Cascade Creek have greater AMS errors of 4.9% (CC12-02) and 8.3% (CC12-05).

We identified three outliers among moraine exposure ages, including samples CB-01,

CB-12 from the Cut Bank terminal and CB-23 from the Cut Bank recessional moraine (Table 1).
Sample CB-01 is more than 8 ka older than all other boulder exposure ages from the terminal



moraine and is therefore interpreted to reflect inherited $^{10}$Be nuclide inventory in the surface from
a period of prior exposure. Sample CB-12 is younger than all but one of the exposure ages on the
upvalley recessional moraine, which is interpreted to represent incomplete or inconsistent
exposure history since the terminal moraine was deposited. Sample CB-23 has an exposure age 3
ka younger than the three other boulders from the moraine and is also interpreted to represent
incomplete or inconsistent exposure history since the recessional moraine was deposited. Although
we found no evidence in the field for inconsistent exposure histories among the sampled boulders,
these young exposure ages could be explained by several geologic processes including local burial
by sediment followed by exhumation, or significant boulder-surface erosion rates. The mean of
the remaining seven $^{10}$Be exposure ages from the terminal moraine in Cut Bank Creek valley limit
its abandonment to 17.2 ± 0.2 ka. The abandonment age of the recessional moraine, 16.4 ± 0.2 ka,
is defined by three exposure ages.

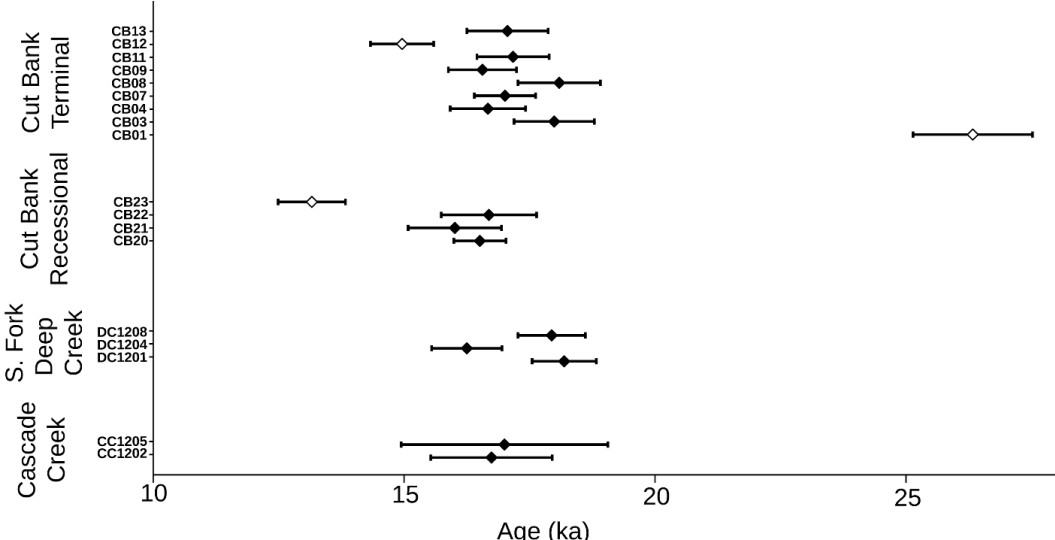


*Figure 4. Cosmogenic $^{10}$Be exposure ages with analytical uncertainties for samples collected from*
*moraines at Cascade Creek and South Fork Deep Creek in the Absaroka Range and moraines at*



*Cut Bank in the Lewis Range. Samples that have been identified as outliers are denoted by open*
*symbols.*

Abandonment ages for the two moraines in the northern Absaroka Range at Cascade

Canyon and South Fork Deep Creek are limited by the means of two and three boulder exposure
ages at $16.9 \pm 0.1$ ka and $17.5 \pm 0.6$ ka, respectively, although we reiterate that the exposure ages
of the lateral moraine at Cascade Canyon are considered preliminary because of the greater-than-
expected AMS measurement errors.

The set of bedrock exposure ages from the ice-recessional path in Pine Creek valley

includes one (PC11-03, $34.0 \pm 2.8$ ka) that exceeds the exposure age of the lateral moraine
downvalley ($18.2 \pm 0.5$ ka, Licciardi and Pierce, 2008) and two (PC11-04 and PC11-10, $18.1 \pm 2.1$
and $18.3 \pm 0.8$ ka, respectively) that overlap with it. These surfaces are interpreted to reflect
inherited $^{10}$Be from a period of prior exposure, which suggests that glacial scouring during the last
glaciation at these sample sites was insufficient to remove the $^{10}$Be inherited from pre-glacial
exposure of the valley floor. Two samples, PC11-07 and PC11-12, yield exposure ages younger
than surfaces sampled at upvalley positions and are interpreted to reflect incomplete exposure due
to burial by sediment. Sample PC11-07 is from an erratic boulder atop a bedrock surface with
exposure ages 3 kyr older, suggesting that the boulder, originally interpreted to be an erratic
deposited by glacier ice during recession, has been reworked by fluvial and mass-movement
processes. The remaining six exposure ages range from $16.0 \pm 0.6$ ka at the farthest downvalley
site (PC11-11) to $13.7 \pm 0.3$ ka at the farthest up valley site (PC11-01) in the cirque occupied by
Pine Creek Lake. When combined with the mean exposure age of the latero-terminal moraine of
$18.2 \pm 0.5$ ka, these exposure ages record the pace and timing of ice retreat over a period of ~4 kyr.





Table 1. Cosmogenic [10]Be sample information and exposure ages

| Sample ID | Latitude | Longitude | Elevation (m) | Thickness (cm) | Density (g cm$^{-2}$) | Shielding Factor | Erosion rate (cm yr$^{-1}$) | [Be-10] (atmos g$^{-1}$) | +/- | Exposure Age (ka) | Analytical Error (ka) | External Error (ka) |
|---|---|---|---|---|---|---|---|---|---|---|---|---|
| *Cut Bank Terminal Moraine* | | | | | | | | | | | | |
| CB-01 | 48.5936 | -113.1555 | 1444 | 1 | 2.65 | 1 | 0 | 3.96E+05 | 1.78E+04 | 26.3 | 1.2 | 1.6 |
| CB-03 | 48.6009 | -113.1577 | 1452 | 3 | 2.65 | 1 | 0 | 2.64E+05 | 1.17E+04 | 18.0 | 0.8 | 1.1 |
| CB-04 | 48.6016 | -113.1565 | 1450 | 3.8 | 2.65 | 1 | 0 | 2.42E+05 | 1.08E+04 | 16.7 | 0.8 | 1.0 |
| CB-07 | 48.6260 | -113.1673 | 1474 | 4.3 | 2.65 | 1 | 0 | 2.51E+05 | 8.93E+03 | 17.0 | 0.6 | 0.9 |
| CB-08 | 48.6257 | -113.1670 | 1472 | 3 | 2.65 | 1 | 0 | 2.70E+05 | 1.21E+04 | 18.1 | 0.8 | 1.1 |
| CB-09 | 48.6210 | -113.1633 | 1465 | 3 | 2.65 | 1 | 0 | 2.45E+05 | 1.00E+04 | 16.6 | 0.7 | 0.9 |
| CB-11 | 48.6196 | -113.1656 | 1469 | 3 | 2.65 | 1 | 0 | 2.55E+05 | 1.07E+04 | 17.2 | 0.7 | 1.0 |
| CB-12 | 48.6345 | -113.1913 | 1523 | 3 | 2.65 | 1 | 0 | 2.31E+05 | 9.61E+03 | 15.0 | 0.6 | 0.8 |
| CB-13 | 48.6338 | -113.1910 | 1518 | 2.8 | 2.65 | 1 | 0 | 2.64E+05 | 1.25E+04 | 17.1 | 0.8 | 1.0 |
| | | | | | | | | **Landform Age (ka)** | | **17.2** | | |
| | | | | | | | | **Standard Error (ka)** | | **0.2** | | |
| *Cut Bank Recessional Moraine* | | | | | | | | | | | | |
| CB-20 | 48.6170 | -113.2336 | 1539 | 4 | 2.65 | 1 | 0 | 2.57E+05 | 8.07E+03 | 16.5 | 0.5 | 0.8 |
| CB-21 | 48.6115 | -113.2447 | 1544 | 4 | 2.65 | 1 | 0 | 2.50E+05 | 1.44E+04 | 16.0 | 0.9 | 1.1 |
| CB-22 | 48.6111 | -113.2418 | 1534 | 2 | 2.65 | 1 | 0 | 2.63E+05 | 1.49E+04 | 16.7 | 1.0 | 1.1 |
| CB-23 | 48.6108 | -113.2414 | 1532 | 5 | 2.65 | 1 | 0 | 2.01E+05 | 1.02E+04 | 13.2 | 0.7 | 0.8 |
| | | | | | | | | **Landform Age (ka)** | | **16.4** | | |
| | | | | | | | | **Standard Error (ka)** | | **0.2** | | |
| *Cascade Creek* | | | | | | | | | | | | |
| CC12-02 | 45.4688 | -110.5449 | 1934 | 4 | 2.65 | 0.995 | 0 | 3.31E+05 | 2.39E+04 | 16.7 | 1.2 | 1.4 |
| CC12-05 | 45.4666 | -110.5428 | 1968 | 4 | 2.65 | 0.995 | 0 | 3.45E+05 | 4.17E+04 | 17.0 | 2.1 | 2.2 |
| | | | | | | | | **Landform Age (ka)** | | **16.9** | | |
| | | | | | | | | **Standard Error (ka)** | | **0.1** | | |
| *S. Fork Deep Creek Lateral Moraine* | | | | | | | | | | | | |
| DC12-01 | 45.5145 | -110.5039 | 2093 | 4 | 2.65 | 0.989 | 0 | 4.05E+05 | 1.41E+04 | 18.2 | 0.6 | 0.9 |
| DC12-04 | 45.5229 | -110.5120 | 1927 | 4 | 2.65 | 0.993 | 0 | 3.19E+05 | 1.37E+04 | 16.3 | 0.7 | 0.9 |
| DC12-08 | 45.5264 | -110.5192 | 1815 | 3 | 2.65 | 0.994 | 0 | 3.28E+05 | 1.22E+04 | 17.9 | 0.7 | 1.0 |
| | | | | | | | | **Landform Age (ka)** | | **17.5** | | |
| | | | | | | | | **Standard Error (ka)** | | **0.6** | | |
| *Pine Creek Bedrock and Erratic* | | | | | | | | | | | | |
| PC11-01 | 45.4840 | -110.4626 | 2761 | 2 | 2.65 | 0.975 | 0 | 4.84E+05 | 1.17E+04 | 13.7 | 0.3 | 0.6 |
| PC11-03 | 45.4859 | -110.4668 | 2774 | 3 | 2.65 | 0.314 | 0 | 4.08E+05 | 3.38E+04 | 34.0 | 2.8 | 3.1 |
| PC11-04 | 45.4862 | -110.4664 | 2768 | 2 | 2.65 | 0.978 | 0 | 6.55E+05 | 7.56E+04 | 18.1 | 2.1 | 2.2 |
| PC11-05 | 45.4885 | -110.4667 | 2752 | 2 | 2.65 | 0.955 | 0 | 5.09E+05 | 2.59E+04 | 14.7 | 0.8 | 0.9 |
| PC11-06 | 45.4885 | -110.4670 | 2757 | 3 | 2.65 | 0.959 | 0 | 5.18E+05 | 1.74E+04 | 15.0 | 0.5 | 0.8 |
| PC11-07 | 45.4886 | -110.4668 | 2765 | 2.5 | 2.65 | 0.956 | 0 | 4.21E+05 | 1.47E+04 | 11.8 | 0.4 | 0.6 |
| PC11-08 | 45.4919 | -110.4772 | 2509 | 3 | 2.65 | 0.918 | 0 | 4.03E+05 | 1.40E+04 | 14.5 | 0.5 | 0.7 |
| PC11-09 | 45.4919 | -110.4773 | 2508 | 3 | 2.65 | 0.918 | 0 | 4.23E+05 | 2.25E+04 | 15.2 | 0.8 | 1.0 |
| PC11-10 | 45.4912 | -110.4873 | 2262 | 2.3 | 2.65 | 0.947 | 0 | 4.48E+05 | 2.02E+04 | 18.3 | 0.8 | 1.1 |
| PC11-11 | 45.4914 | -110.4870 | 2276 | 3 | 2.65 | 0.940 | 0 | 3.88E+05 | 1.33E+04 | 16.0 | 0.6 | 0.8 |
| PC11-12 | 45.4898 | -110.4939 | 2110 | 3 | 2.65 | 0.952 | 0 | 2.95E+05 | 1.03E+04 | 13.7 | 0.5 | 0.7 |

Underlined indicates outlier samples
**Glacier Climate Reconstructions**
Model simulations were completed for the Cut Bank and northern Absaroka model
domains including four simulations matching the: Cut Bank terminal moraine (CB$_T$), Cut Bank
recessional moraine (CB$_R$), and Pine Creek and South Fork Deep Creek lateral sectors of terminal
moraines (NA$_T$; Figure 5). For simplicity, each of the sets of four simulations pin precipitation





change ($P_x$) to multiplicative factors of 0.5, 1.0, 2.0, and 3.0 times modern precipitation, while
temperature depressions ($T_d$) were independently varied in each experiment to match mapped ice
extents. In each of the 12 experiments, calculated ice extents and thicknesses matched well with
field evidence. The twelve experiments define 3 curves (Figure 6), in $T_d$-$P_x$ space, representing
paleoclimate estimates for ice matching $CB_T$ ($R^2 = 0.98$), $CB_R$ ($R^2 = 0.99$) $NA_T$ ($R^2 = 0.99$) with
equations:

(4) $CB_T$    $P_x = 24.084e^{0.3589T_d}$

(5) $CB_R$    $P_x = 6.3721e^{0.2417T_d}$

(6) $NA_T$    $P_x = 16.877e^{0.3379T_d}$



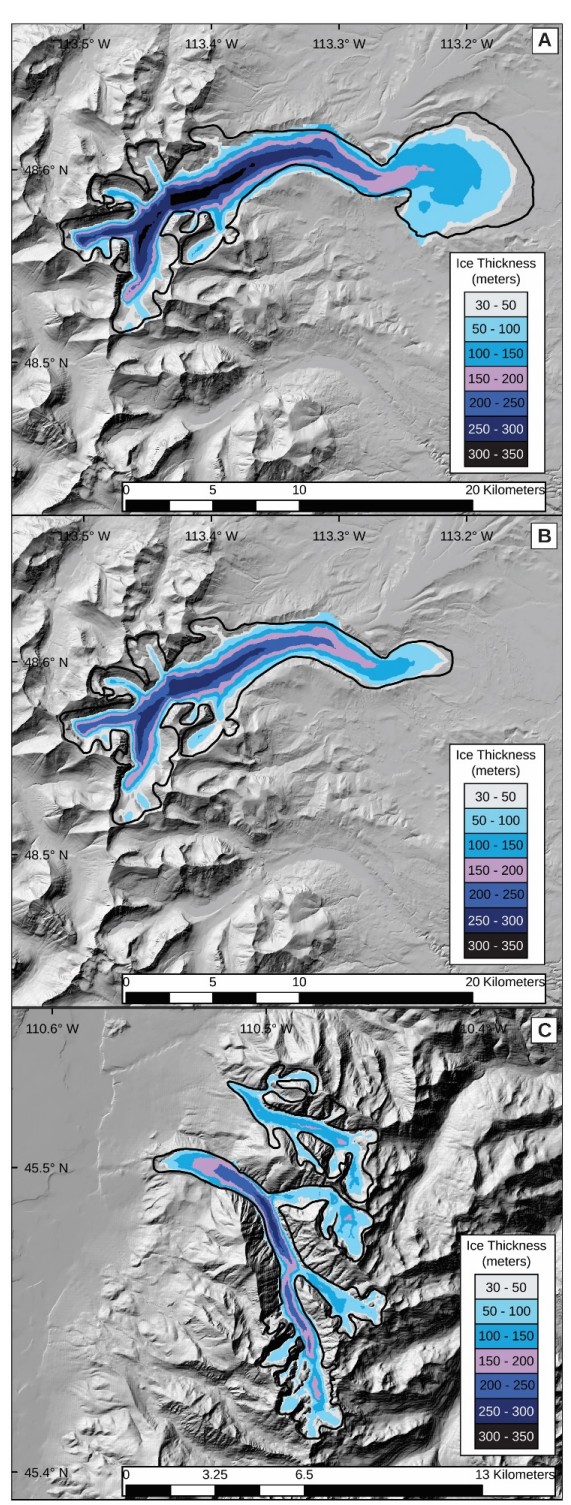


*Figure 5. Ice thickness maps generated*





*from coupled energy-mass balance and ice-flow modeling for A) Cut Bank terminal B) Cut Bank*

*recessional and C) Pine Creek and South Fork Deep Creek in the Absaroka Range. Modeled ice-*

*extents were matched to field evidence (black outlines) by varying precipitation ($P_x$) and*

*temperature ($T_d$) by multiplicative and additive changes, respectively. For each simulation, we*

*found a series of $P_x$-$T_d$ combinations that produced modeled ice extents that satisfactorily matched*

*field evidence.Ice extents shown here use 100% modern precipitation and temperature depressions*

*of -9.2, -8.0, and -8.5 for the Cut Bank terminal (A), Cut Bank recessional (B), and Absaroka*

*Range model domains, respectively.*

As previously mentioned, in order to match the modeled glacier shape to field evidence at Cut

Bank, we found it necessary to effectively set the contribution of ice velocity due to sliding to zero.

In order to test how realistic these model conditions were for reconstructing other glaciers, we

reconstructed the Pinedale glacier that occupied Lake Creek Canyon, the drainage immediately to

the south of Cut Bank. We matched the modeled glacier to the mapped Pinedale maximum in Lake

Creek Canyon with $T_d$-$P_x$ combinations of -8°C & 100% and -6°C & 190% (Supplemental Figure

1). These $T_d$-$P_x$ combinations are both approximately 1°C warmer than results for Cut Bank's

Pinedale maximum glacier given then same precipitation change. However, we find this

compelling evidence that the ice flow parameters we used to reconstruct the Cut Bank Pinedale

glacier are reasonable. The temperature discrepancy between the two sites could be 1) a result

temporal offset between the two maxima as we do not have a landform age for the Lake Creek

terminal moraine 2) a real climatic difference between the two catchments and/or 3) a reflection

of unaccounted for modeling error. With regards to the latter, we assume model uncertainties

matching those reported in Quirk et al. (2020) of ± 1.0°C and 30% for temperature and





precipitation respectively – which indicate overall agreement between the Cut Bank and Lake
Creek simulations.
**DISCUSSION**
**Cosmogenic exposure ages of moraines in a regional and global context**

The $^{10}$Be exposure ages presented here for the South Fork Deep Creek (17.5 ± 0.6 ka) and

Cascade Creek (16.9 ± 0.1 ka) lateral moraines in the northern Absaroka Range are slightly
younger than ages from the previously dated lateral moraine in the neighboring Pine Creek valley
in the northern Absaroka ($^{10}$Be exposure age = 18.2 ± 0.5 ka, with the standard error of ages
recalculated from Licciardi and Pierce, 2008). Although these moraines were deposited by discrete
valley glaciers, their exposure ages are similar to $^{10}$Be exposure age of the nearby Eightmile
terminal moraine (17.9 ± 0.4 ka, recalculated from Licciardi and Pierce, 2008), the outermost
moraine of the last glaciation deposited by the northern outlet glacier of the Yellowstone Icecap,
as well as to the age of the Chico moraine (17.1 ± 0.6 ka recalculated from Licciardi and Pierce,
2008) the initial moraine deposited during recession of this outlet glacier. These ages for outermost
and initial recessional moraine northern Yellowstone/northern Absaroka Range area in
southwestern Montana are also very similar to those we report here for the terminal (17.2 ± 0.2 ka)
and initial recessional (16.4 ± 0.2 ka) moraines at Cut Bank Creek in northwestern Montana. Taken
together, these ages suggest that terminal moraines in western Montana were occupied until ca.
18-17 ka and that glaciers were still near their maximum lengths at ca. 17-16 ka in northern
Yellowstone and in the Lewis Range, as indicated by exposure ages of the recessional moraines.

Moraines in the northern Absaroka Range have exposure ages that fall within the middle

Pinedale interval, 18-16 ka, as identified in the greater Yellowstone region by Licciardi and Pierce
(2018) and after the end of the global LGM (Clark et al., 2009). During this time, the Yellowstone





glacier system thickened across the Yellowstone Plateau, coalesced with ice masses in some
neighboring mountains (such as the Beartooth, High Absaroka, and Gallatin Ranges), and formed
large outlet lobes, including the northern outlet that terminated just south of the glaciated portion
of the Northern Absaroka Range (Licciardi and Pierce, 2008, 2018). This large glacier system
persisted after the southwestern margin of the Laurentide Ice Sheet in northern Montana began
retreating (Dalton et al., 2020) and middle latitudes in the northern hemisphere began warming
(Shakun et al., 2015). Licciardi and Pierce (2018) suggest that enhanced westerly airflow into the
region during the middle Pinedale interval combined with orographic effects of the thickened ice
cap augmented precipitation in the northern Yellowstone region. The strengthened westerly
airflow across the region likely impacted valley glaciers in the northern Absaroka Range,
providing sufficient moisture for glaciers to persist at their maximum lengths despite rising
summer insolation at middle latitudes (Laskar et al., 2004) and atmospheric carbon dioxide
concentrations (Luthi et al., 2008). Additionally, middle latitudes in North America may have
remained cold for several millennia after the Laurentide Ice Sheet began retreating, as suggested
by the persistence of other Rocky Mountain glaciers at near-maximum extents until 17 ka (Laabs
et al., 2020) and model-based estimates of the regional temperatures at 17 ka (Liu et al., 2009; He,

2011).

The terminal and recessional moraines in the Lewis Range have exposure ages that also
fall within the middle Pinedale interval of 18-16 ka and thus may also have been responding to
similar climatic controls as in the Absaroka Range to the south. Alternatively, the post-LGM age
of these moraines could be related to the Lewis Range's proximity to the southwestern margin of
the Laurentide Ice Sheet. When the Shelby Lobe and other southwestern outlets of the Laurentide
Ice Sheet were at their maximum extent, general circulation modeling studies suggest that a large
area of high atmospheric pressure developed across the western dome of ice sheet resulting in
anticyclonic, easterly airflow along the southern margins (Thompson et al., 1993; Bartlein et al.,
1998). This circulation pattern likely resulted in cold and dry climate in the Lewis Range while
the southwestern outlets occupied their terminal moraines. Recent reconstructions of this sector of
the Laurentide Ice Sheet suggest that the Shelby Lobe retreated to the northeast by ca. 17 ka
(Dalton et al., 2020), which may have been accompanied by a weakening of easterly, anticyclonic
circulation at the latitude of the Lewis Range and strengthening westerly airflow that delivered
moisture-laden air and enhanced precipitation in the mountains. Enhanced precipitation may have
resulted in glacier advance to their maximum lengths after the Laurentide Ice Sheet began to
retreat. This effect has been suggested by previous studies, including earlier interpretations of the
moraine chronologies in northern Yellowstone region (Licciardi et al., 2001) and age limits on
moraines elsewhere in northern interior mountains (Licciardi et al., 2004; Thackray et al., 2004).
Licciardi and Pierce (2018) note that the range of terminal-moraine exposure ages in the
Yellowstone region includes some that overlap with the early Pinedale interval of 22-18 ka, which
includes the latter part of the global Last Glacial Maximum when some southwestern outlets of
the Laurentide Ice Sheet were at their maximum size. While the effect of the southwestern
Laurentide on regional airflow may not have impacted the Yellowstone region, it may have
impacted the Lewis Range as indicated by the exposure ages of the terminal and recessional
moraines in Cut Bank valley. Additional age limits on moraines in the Lewis Range, other
mountains in northwestern Montana, and the Shelby Lobe will aid in understanding the relative
timing of mountain and continental glaciation.

Considering the glacial chronologies presented here in a larger spatial context, the exposure

ages of terminal and recessional moraines show some consistency with mountain glacier moraines




from elsewhere in the western United States. Elsewhere in the Rocky Mountains, moraines with
age limits of ca. 18-17 ka are found in the Sawtooth Range in Idaho (Thackray et al., 2004), the
Wasatch and Uinta Mountains in northern Utah, and numerous glacial valleys in the Southern
Rocky Mountains in Colorado (Leonard et al., 2017a; Brugger et al., 2018, 2019; Schweinsberg et
al., 2020). Where sequences of moraines are exposure-dated in the Rocky Mountains, the
outermost moraines of the last glaciation generally have ages that fall within the early Pinedale
interval of 22-18 ka and inner moraines (representing near-maximum glacier lengths) that fall
within the middle Pinedale interval (Quirk et al., 2020; Laabs et al., 2020). This pattern is observed
throughout the Rocky Mountains and suggests that the mountain glacier moraine chronology in
western Montana differs from the rest of the region, such that the outermost moraines do not
represent the early Pinedale interval and only represent the middle Pinedale interval. This may
reflect the importance of regional climatic effects on mountain glaciation, especially the
strengthening of westerly airflow and attendant moisture delivery, as described above.
**Inferred paleoclimate for the last glaciation**

Glacier modeling results yielded a series of $P_x$-$T_d$ combinations that produced ice extents

that closely matched mapping-based reconstructions of the for the Cut Bank terminal and
recessional positions, and for the terminal positions in the Pine Creek and South Fork Deep Creek
valleys (Figure 6). Our results, particularly at Cut Bank, broadly agree with previous inferences of
regional Late Pleistocene climate, including pollen-based reconstructions and other applications
of paleoglaciology (Mumma et al., 2012; Murray and Locke, 1989; Locke, 1990; Birkel et al.,
2012). However, to infer changes in precipitation or temperature from our glacier modeling, one
of the two variables must be limited independently (i.e., from relevant paleoclimate proxy records).



In the following paragraphs, we consider the modeling results in the context of some existing
inferences of paleoclimate based on other proxy records in western Montana.

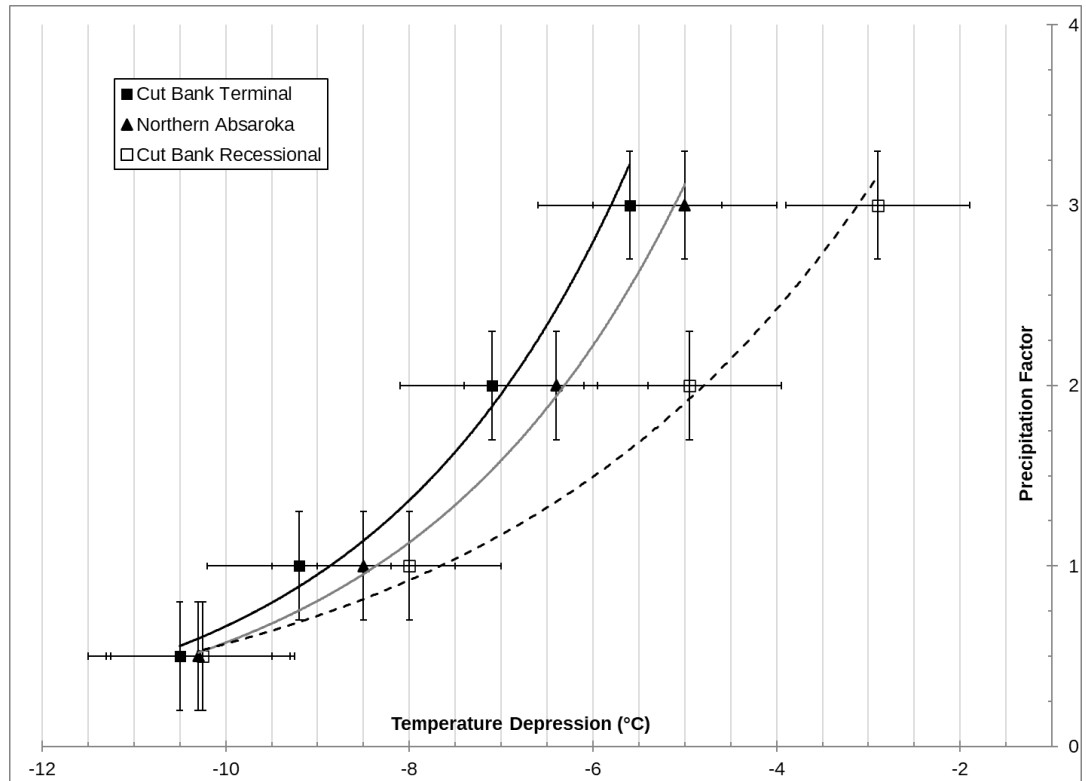


*Figure 6. Multiplicative precipitation factors and temperature depressions, both with respect to*

*modern, that produced modeled ice extents matching field mapped extents for 1) Cut Bank terminal*

*moraine 2) northern Absaroka Range Pinedale maxima at Pine Creek and South Fork Deep Creek*

*and 3) Cut Bank recessional moraine.*

Mumma et al. (2012) presented a paleoclimate record developed from Lower Red Rock

Lake in southwestern Montana, alongside a synthesis of other lacustrine records from the region,
spanning approximately from the entire LGM time interval (i.e. 26-19 ka) through the early
Holocene. The Lower Red Rock Lake chronology is constrained by several [14]C ages from organic
sediments and wood, plant, and peat material as well as tephrochronology. Recalibration of





radiocarbon ages using IntCal20 (Reimer et al., 2020) calibration data results in changes to the
ages of ≤ 3% and therefore does not change the interpretations presented by Mumma et al. (2012).
Their interpretations of the pollen and sedimentological records indicate that from ca. 28-17 ka,
southwestern Montana was dominated by a cold and dry climate. During the subsequent interval
of 17.0-10.5 ka, coinciding with regional deglaciation, they suggest that temperatures increased
relative to the 28-17 ka period of their record but were still colder than modern and that effective
moisture likely increased. Mumma et al. (2012) attribute the rise in precipitation beginning at 17.0
ka to a northward migration of the jet stream and increasing summer insolation. Such a shift in
climate at 17 ka may be reflected in the glacial chronology presented here. Increased precipitation
commencing at 17 ka may have augmented the mass balance of mountain glaciers resulting in ice
advance to the terminal moraines. A glacier response to increased regional precipitation is
consistent with the assertion that increased westerly airflow accompanied glacier growth in the
Yellowstone region during the middle Pinedale interval. Alternatively, if cold and dry climate
during the interval 28-17 ka favored mountain glacier maxima, then the shift to warmer and wetter
climate at 17 ka may have initiated ice retreat from terminal moraines.

Reconstructions of the valley glacier that occupied Big Timber Canyon in the Crazy

Mountains of western Montana by Murray and Locke (1989) provide additional limits for regional
climate during the last glacial culmination. Their glacier model experiments, specifically the low
mass-balance gradients derived from them, indicate that climate in the northeastern Crazy
Mountains was typical of a cold, dry continental interior, with around 75% of modern precipitation,
when the glacier reached its maximum size, although the specific timing of the glacier maximum
here is unknown. Additional work by Locke (1990, 1995) on paleoglacier reconstructions suggests
that last-glaciation ELAs were ~450 m lower but followed a parallel trend to those of modern



glacier ELAs, which he interprets to indicate similarities in temperature distributions and westerly
airflow across the northern Rocky Mountains of western Montana. By using the difference
between modern and Pleistocene ELAs to compute precipitation during the local glacial maximum
(for an assumed temperature depression of 10°C), Locke (1990) found that accumulation-season
precipitation ranged from 50 cm less than modern to 50 cm greater than modern (in units of water
equivalent) across mountain ranges in western Montana. However, Locke suggested that based on
the overall pattern of ELA change that climate in western Montana was likely drier during the
LGM. If precipitation changes during the last glacial culmination at 18-17 ka was 75% of modern
then our modeling results suggest the accompanying temperature depression in the northern
Absaroka Range and in the Lewis Range was around 8-10°C. The magnitude of regional climate
change at 18-17 ka in the Crazy Mountains is unclear, however, and may have differed between
the latitudes of glacial valleys in the Lewis Range (N48.6°) and that of glacial valleys in the
northern Absaroka (N45.5°).

While a unique temperature-precipitation combination for the culmination of the Pinedale

maximum in western Montana is difficult to infer from glacier modeling results presented here,
the consistent timing of the glacial culmination at 18-17 ka – after the Laurentide Ice Sheet began
retreating and global LGM – suggests that a regional increase in precipitation during the middle
Pinedale interval supported glacier maxima. This is consistent with inferred climate for the last
glaciation in the greater Yellowstone region described by Licciardi and Pierce (2018) and earlier
studies inferring that glaciers in northern mountains in the conterminous western United States
reached the maximum size after the Laurentide Ice Sheet began retreating (Thackray, 2008), as
well as the regional airflow pattern implied by the paleoglacier reconstructions of Locke (1990,
1995) and pollen records reported by Mumma et al. (2012). If strengthened westerly airflow at 18-





17 ka resulted in accumulation-season precipitation similar to modern amounts as suggested by
regional climate proxies and model output, then a regional temperature depression can be inferred
from glacier modeling results presented here. Model simulations of glaciers in the Pine Creek and
South Fork Deep Creek valleys suggest a temperature depression of 8.5° ± 1.0°C in southwestern
Montana, whereas model simulations of the glacier in Cut Bank Creek valley suggest a temperature
depression of 9.2° ± 1.0°C in northwestern Montana. This magnitude of cooling for the last glacial
culmination in western Montana is consistent with output of some general circulation models
involved in the Paleoclimate Model Intercomparison Project (PMIP3), although these results
represent climate at 21 ka while the Laurentide Ice Sheet was still present in western Montana.
Specifically, the average temperature change predicted for western Montana by all PMIP3
ensembles is -12.9 ± 4.9 °C (1-sigma; interpolated by Oster et al., 2015).
**The pace of ice retreat in the Rocky Mountains**

Ice-margin retreat rates following the abandonment of Pinedale maximum extents in the

northern Rockies are constrained by the cosmogenic exposure age chronology of glacially scoured
and striated bedrock from Pine Creek Canyon in the northern Absaroka Range (Figure 2). First,
we emphasize the uncertainty associated with this deglacial chronology from the exclusion of three
assumed old and one young outlier from the data set. Furthermore, the sample transect only
captures a northern tributary of the Pine Creek glacier (see sample transect in Figure 2) and thus
may not be representative of the larger main-valley glacier system. However, few glaciated valleys
in the northern Rockies have age controls sufficient to estimate retreat rates, therefore the data
presented here, while limited, are valuable for inferring rates of deglaciation. Keeping these
considerations in mind we can use the data to describe the pattern of deglaciation in the northern
Absaroka.



We model the Pine Creek glacier retreat rates using linear regressions of all or select
subsets of the age and sample distance or elevation data (Figure 7). The models indicate horizontal
ice-margin retreat rates ranging from 1.0 km ka$^{-1}$ to approximately 2.6 km ka$^{-1}$ and vertical retreat
ranging from 205 to 288 m ka$^{-1}$.  The data also suggest that the main body of ice in the Pine Creek
glacier had separated from the northern tributary by ca. 16 ka, and by 13.7 ka, the northern tributary
had undergone an 80% reduction in length and retreated over 1.1 km in elevation from the terminal
moraine. The remaining deglacial history of the Pine Creek glacier following the inferred recession
around 13.7 ka is not constrained by the cosmogenic chronology reported here.
Several studies of glaciated valleys in the western U.S. have sufficient age controls to
estimate retreat rates during the last glaciation along a north-south transect of the Rocky Mountains
including (Figure 8) the Pine Creek valley reported here, the Teton Range in Wyoming (Licciardi
and Pierce, 2008), the Wasatch Range and Uinta Mountains in northern Utah (Laabs et al., 2011;
Munroe and Laabs, 2017; Quirk et al., 2018, 2020), the Front Range (Ward et al., 2009; Duhnforth
et al., 2011), Sawatch Range (Briner, 2009; Young et al., 2011; Leonard et al., 2017b;
Schweinsberg et al., 2020; Tulenko et al., 2020), and San Juan Range (Guido et al., 2007) in
Colorado. Here, we consider vertical retreat rates for all sites to minimize the strong effects valley
slope and glacier hypsometry have on apparent rates of retreat.
Vertical glacier retreat rates exhibit no clear relationship with respect to latitude along a
north-south transect from Pine Creek in southern Montana to the San Juan Rane in Colorado.
Retreat rates for sites in the middle of the transect (Wasatch, Uinta, Front Range) are somewhat
lower than rates calculated from the remaining sites and could reflect a response to increased
moisture at these latitudes during Heinrich Stadialsl 1 (e.g. Munroe and Laabs, 2013). While the
timing of initial abandonment of ice-distal positions is variable across the Rockies, ranging from





the end of the LGM to ca. 16 ka, the broad pattern and timing of subsequent glacier retreat is
similar across the Rocky Mountains (Figure 8)



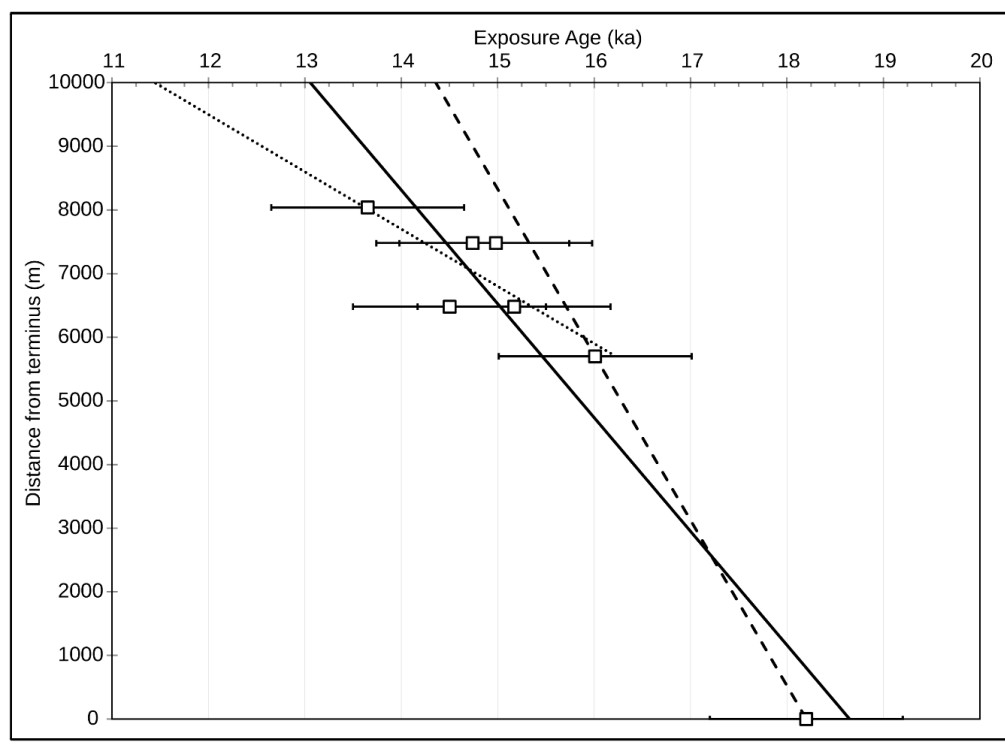

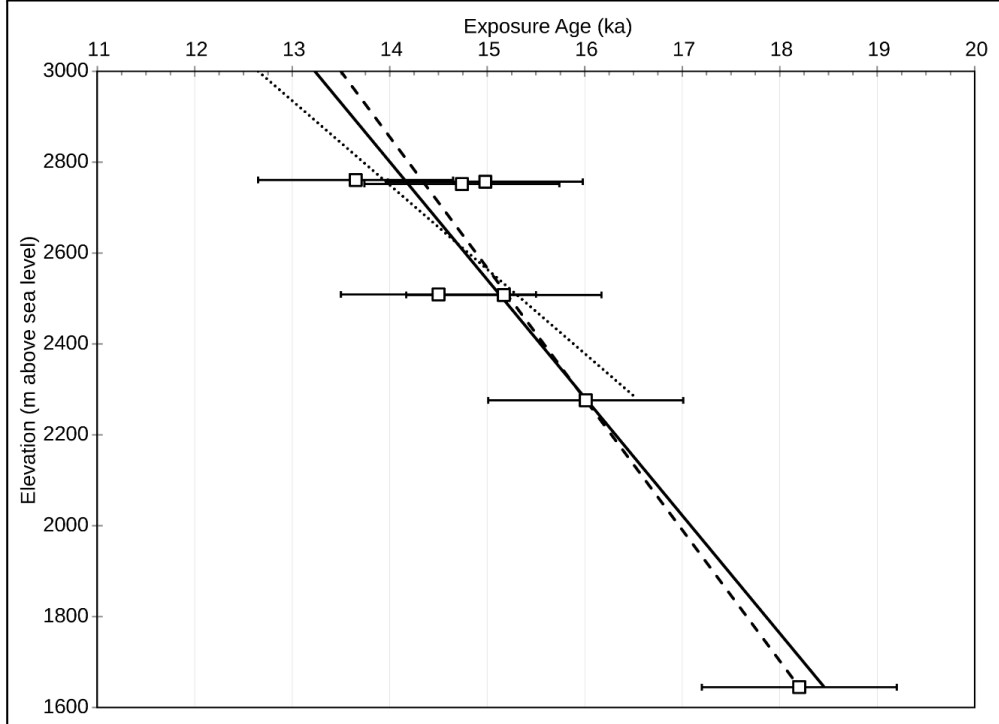



*Figure 7. Time-distance (top) and time-elevation (bottom) diagrams illustrating retreat from the*

*Pine Creek Pinedale maximum extent (18.2 ± 1.3 ka; Licciardi & Pierce, 2008) from cosmogenic*

*exposure ages collected from striated and polished bedrock along a longitudinal transect of Pine*

*Creek. For both plots, three different retreat rates are indicated by best fit linear curves through*

*all of the data (solid line), lateral moraine and nearest bedrock sample (dashed line), and only*

*bedrock ages collected in this study (dotted line). All curves are extrapolated to 10 km (top) and*

*3000 m (bottom), which are the approximate distance to and elevation of the headwall.*

The timing of terminal moraine abandonment is variable across the Rocky Mountains and

span a period of around 8 ka, beginning during the LGM and continuing, such as in the Lewis and

Absaroka ranges, into the middle Pinedale (18-16 ka). The large range in glacier retreat from ice-

distal positions suggests diverse controlling mechanisms of initial deglaciation across the region.

However, the coherence of ice-retreat rates in the Absaroka Range with locations across the

Rockies from ca. 16 ka through the Lateglacial suggests common factors driving deglaciation

across the region. For example, glacier retreat in Rocky Mountains after ca. 16 ka coincides with

sustained increases in atmospheric $CO_2$ and regional temperature changes despite some glacier

retreat lagging behind initial rises in $CO_2$ around 17 ka (Figure 8). Alternatively or in addition,

modeling studies have highlighted the effect North American ice-sheets, and in particular their

demise, have on regional climate (Lora et al., 2016; Tulenko et al., 2020). Specifically, the

separation of the Laurentide and Cordilleran ice sheets around 15-16 ka (Dalton et al., 2020) may

have led to drier and warmer conditions across Western North America (Lora et al., 2016) and thus

may have contributed to sustained glacier retreat observed in the Rocky Mountains during this

time period (Figure 8). Whatever the mechanism, the data presented here highlight the dramatic



age-range of initial terminal moraine abandonment and regional coherence of sustained glacier

retreat throughout the Lateglacial.

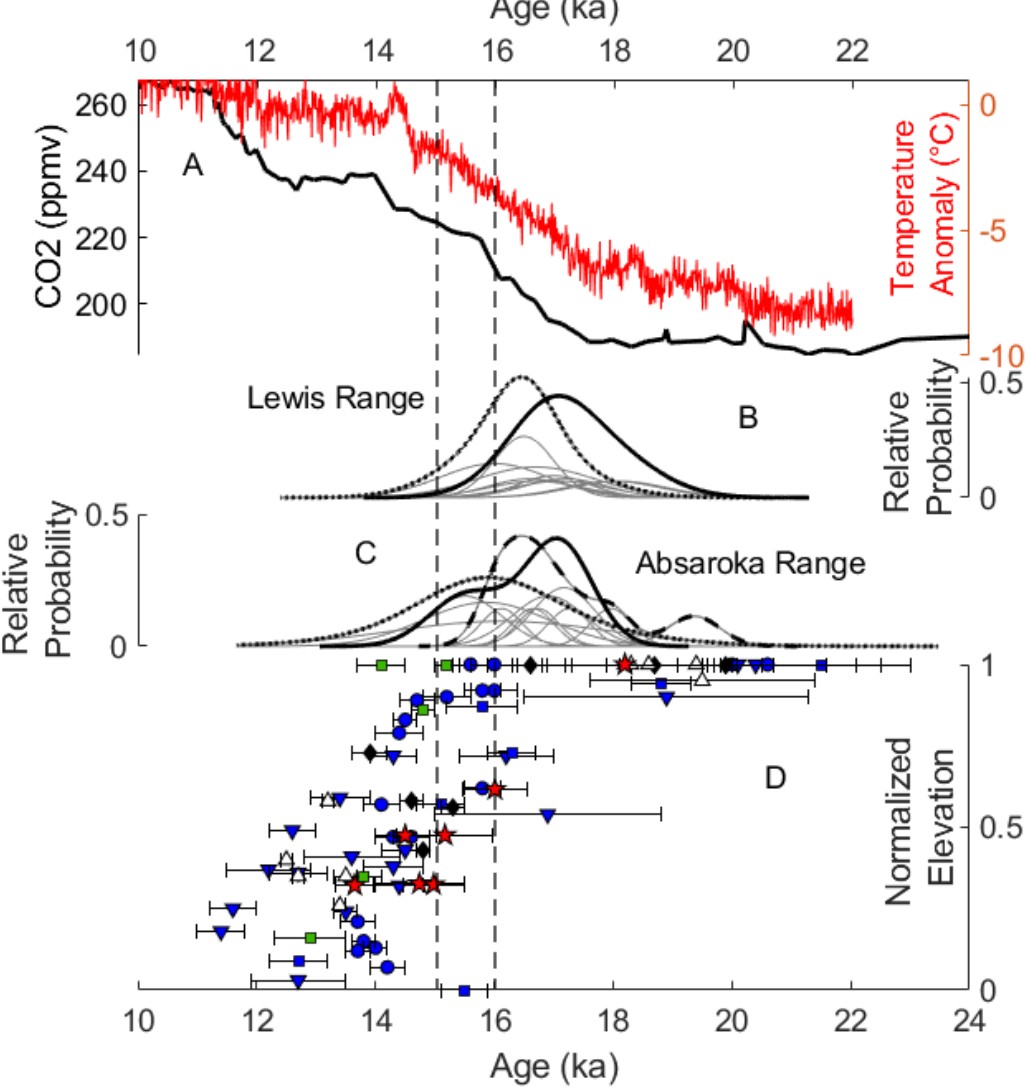

*Figure 8. (A) Surface temperature anomalies from TRACE-21ka for Western North America (red)*

*and Epica-Vostok composite CO₂ concentrations. (B) Camel plots of exposure age data from Cut*

*Bank glacier in the Lewis Range: Cut Bank terminal (solid line), Cut Bank recessional dotted line)*





*and (C) Absaroka Range: South Fork Deep Creek (solid line), Cascade Creek (dotted line), and*

*Pine Creek (dashed line). (D) Normalized glacier elevation for Pine Creek glacier in the Absaroka*

*Range (red stars), Teton Range, WY (green boxes), Wasatch Range, UT (black diamonds), Uinta*

*Mountains, UT (white triangles), Front Range, CO (blue inverted triangles), Sawatch Range, CO*

*(blue circles), and San Juan Range, CO (blue squares). Dashed vertical lines bracket the*

*approximate timing of the separation of the Laurentide and Cordilleran ice-sheets in North*

*America.*

**CONCLUSIONS**

We present cosmogenic exposure ages for moraines in the Absaroka and Lewis Ranges

of Montana that indicate glacial stadials during the middle Pinedale interval (18-16 ka) and thus

after the end of the LGM. We propose that regionally strengthened westerly airflow and

orographic effects associated with the thickening Yellowstone Ice Cap nourished valley glaciers

in the Absaroka Range with precipitation and allowed glaciers to persist at their maximum

lengths despite rising summer insolation at middle latitudes (Laskar et al., 2004) and rising

atmospheric carbon dioxide concentrations (Luthi et al., 2008). Similarly in the Lewis Range,

glaciers maintained their maximum extents following the retreat of the Shelby Lobe of the

Laurentide Ice Sheet by ca. 17 ka (Dalton et al., 2020), which we propose could have been

accompanied by a weakening of anticyclonic circulation and strengthening of westerly airflow

that effectively increased precipitation in the Lewis Range. If we assume that precipitation

during the middle Pinedale was similar to or slightly drier than modern, following a cold and

likely much drier than modern early Pinedale / LGM, our model simulations of glaciers in the

Absaroka Range suggest a temperature depression around 8.5-9.0°C ± 1.0°C, while model

simulations of the Cut Bank glacier in the Lewis Range suggest a temperature depression around





706 9.0-10.0± 1.0°C. Ice-retreat rates from Pine Creek Valley in the Absaroka Range likely ranged

707 from 1.0 to 2.6 km/ka and vertical retreat ranging from 205 to288 m ka$^{-1}$ and broadly coincide

708 with other Rocky Mountain records of glacier retreat.

709 **CODE & DATA AVAILABILITY**

710  Cosmogenic $^{10}$Be exposure age sample, AMS, and chemistry data are available in Table 1

711 and Supplemental Table 1. Glacier energy-mass balance and ice-flow model code available upon

712 request.

713 **COMPETING INTERESTS**

714  The authors declare that they have no conflict of interest.

715 **AUTHOR CONTRIBUTIONS**

716  Brendon Quirk, Elizabeth Huss, Benjamin Laabs, and Eric Leonard conceived the project

717 with input from Joseph Licciardi, and Mitchell Plummer. All listed co-authors completed field

718 mapping and sampling. Authors Quirk, Huss, and Laabs completed prep work for $^{10}$Be exposure

719 dating. Marc Caffee assisted with measurement of $^{10}$Be/$^9$Be ratios. Brendon Quirk completed all

720 glacier modeling with significant input from the other authors. All authors contributed to data and

721 modeling interpretations. Brendon Quirk and Benjamin Laabs wrote the manuscript.

722 **ACKNOWLEDGMENTS**

723  We thank the reviewers of this manuscript in advance. We also thank Doug Steen and

724 Alec Spears for help sampling the South Fork Deep Creek and Cascade Creek moraines and with

725 preliminary glacier model results.



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
