# Peer review of "Late Pleistocene glacial chronologies and paleoclimate in the"

_Climate of the Past, 2021_

## Author Comment (AC1)

Clim. Past Discuss., referee comment RC2
https://doi.org/10.5194/cp-2021-106-RC2, 2021

[Figure]

**Comment on cp-2021-106**
Anonymous Referee #2

Referee comment on "Late Pleistocene glacial chronologies and paleoclimate in thenorthern Rocky Mountains" by Brendon Quirk et al., Clim. Past Discuss., https://doi.org/10.5194/cp-2021-106-RC2, 2021

This manuscript presents valuable new data from the northwestern USA from an area where more study has been needed. Furthermore, it also provides valuable and important modeling simulations (constrained by the evidence) that allow authors to infer possible or likely temperatures and precipitations that expanded the glaciers that formed the glacial deposits. Having both in the same paper is done quite elegantly by the authors. The authors then use their findings to discuss possible changing climates south of the Laurentide and Cordilleran at the end of the last deglaciation.

Despite my numerous comments, and some items I think need to be revised, I enjoyed reading the paper and look forward to seeing it in print eventually. I would put my rating as moderate revisions, as none of them should be that difficult for the authors to address. My most important comments probably are how the cosmogenic nuclide data are calculated and presented (see below). This may affect the interpretations; not in a major way, but enough that they should be addressed. Also, see other major comments below (i.e., not just clarifying or typos).

In general, it is well organized and written, but there are some awkward and even at times unclear sentences, a few grammar problems, and a few typos, that are easy enough to fix (below) before publication

Lines 20-21 – near former ice sheet margins?

Yes that is correct. We have updated the text as follows: "However, relatively few glacial records have been developed for mountain glaciers in the northern Rocky Mountains near **former** ice sheet margins"

Line 22. results, which show?

We have updated the text as follows: "Here, we report cosmogenic beryllium-10 surface exposure ages and numerical glacier modeling results **which show** that mountain glaciers"

22-25. This sentence is awkward or confusing as it seems two parts? One is explaining conditions that caused LGM (late or peak?) I think, other part of sentence is timing for end of LGM? I suggest two sentences.

We appreciate the reviewer's comments. However, we have declined to expand this section into sentences in the interest of brevity.

41-43..Pleistocene…Pleistocene… (awkward writing)…cosmogenic nuclide chronologies..

We have updated the text as follows: "While surficial geologic records of Pleistocene mountain glaciation in the northern Rocky Mountains of western Montana have been available for decades (Alden, 1932; Carrara, 1987), these records have seldom been used to infer climate conditions (e.g., Murray and Locke, 1989)"

56. in some areas, they coalesced….

We are unclear as to what this comment is suggesting.

90-91 – how do you know 200 m thick and in excess of 300 m? can you add one more sentence saying how derived (lateral moraines, other?). Or give a reference?

We have updated the text as follows: "The glacier was over 25 km long at its maximum extent and in many areas was over 200 meters thick with maximum ice thickness in excess of 300 meters as evidenced from trimline elevations"

100-101 – same question as 90-91. Just mention briefly or give a reference(?).

We have updated the text as follows: "Ice thicknesses were thinner in the Absaroka Range glaciers as compared to the Cut Bank glacier, with many areas hosting 100-200 meter thick ice and maximum thicknesses in Pine Creek of 250-300 meters (Licciardi et al., 2001; Licciardi and Pierce 2008)."

126. If no numerical ages, how did Fullerton "identify" a bull lake and pre bull lake? Maybe say Fullertin inferred?

Yes, we think this probably more appropriate phrasing. We have updated the text as follows: "Fullerton et al. (2004) inferred multiple Pinedale tills, two ages of Bull Lake till, and a possible pre-Bull Lake till in moraine deposits at Cut Bank Creek and elsewhere along the eastern front of the Lewis Range"

128. please give a 1s or 2s calibrated (for 14C) age range here, or the uncertainty that the online calibration software gives.

Thank you for pointing this out. We have recalibrated the age and updated the text as follows: "No numerical ages are available for these deposits, although a radiocarbon age on a wood fragment, underlying two latest Pleistocene tephra layers in lake sediment at Marias Pass, provides a minimum age of 12,194±145 14C yr (Carrara, 1995) or 13.8-14.8 cal yr (Fullerton et

al., 2004; recalibrated here using IntCal13 (1σ); Reimer et al., 2013) for complete recession of at least one east-side outlet glacier of the Northern Montana Ice Cap in the Glacier National Park region."

136-137. Some of this is albeit beyond this manuscript to address. But. I think the way this is phrased is slightly awkward. I do not fully understand their comparison to the global LGM as discussed. The timing of maximum extent of the LGM sensu stricto, that is the coldest part, may have varied slightly around the world. And, although maximum limits sensu stricto were maybe before 19 ka (Clark et al), there was still a lot of ice around until at least 18 ka and even later in places, as shown here. I think there is a bit of apples and bananas being compared in the 136-137 line. Yes, middle Pinedales were not as extensive as maximum Pinedale extent (global LGM?) by definition. Does it mean still local LGM conditions? Not 'clearly postdated.'

A point is, how do the authors interpret or see the global LGM as defined in Clark et al.? In terms of how Clark et al define the global LGM (ice sheets, Volume, Sea level?) and do the authors agree it is apples and apples to compare their findings? Simply saying younger than Clark et al LGm does not give reader context for how Clark et al (2009) define it , and what this paper shows? Maybe the authors can clarify.

Here, we use the LGM definition from Clark et al. (2009), a time period (26.5-19.0 ka) of global ice volume maximums (defined in the Previous Studies section), as a relative temporal comparison for mountain glacier chronologies. We agree that it many mountain glaciers occupied terminal positions after the global LGM – a point that our data demonstrate but we do not take for granted as other data from WNA clearly show moraine abandonment during the LGM (e.g. some canyons in the Wasatch Range of Utah).

143- this text or other paper's text?

Yes, we agree that as written the text is ambiguous. We have updated the text as follows: "Terminal and recessional moraines at the southwestern front of the northern Absaroka Range and in the neighboring Paradise Valley to the south have cosmogenic 10Be exposure ages that were originally reported by Licciardi et al. (2001) and combined with additional data by Licciardi and Pierce (2008; 2018). The terminal moraine in Pine Creek valley of the northern Absaroka Range has a mean cosmogenic 10Be exposure age of 18.2 ± 0.5 ka (± 1 standard error of the mean; recalculated using methods described in the Methods section) "

148-149. same comment. I am not sure what it means to say 'critically after the end of the global LGM' How is the global LGM defined compared with their mountain glaciers? If all rocky mountain recorded being discussed here go to 18 ka-17ka, maybe that is the end of the local LGM?

Yes, we agree with the reviewer here in that we are simply making a temporal comparison to a time period of maximum global ice volumes. The use of the word 'critically' is unnecessary and unhelpful and we have removed it from the sentence – indicating a basic temporal comparison.

157. They interpreted…

We have updated the text to reflect this edit.

166-167 – awkward writing.

Yes, we very much agree! The text has been updated for clarity as follows: "Modern methods used to reconstruct paleo-glaciers, particularly distributed energy/mass-balance or degree-day mass-balance models, have been successfully applied to sites in the Middle (Laabs et al., 2006; Refsnider et al., 2008; Birkel et al., 2012; Quirk et al., 2018, 2020) and Southern Rocky Mountains (Ward et al., 2009; Brugger, 2010; Brugger et al. 2018, 2019; Dühnforth and Anderson, 2011; Leonard et al., 2014, 2017a; Schweinsberg et al., 2016)"

183. …on their crests?

Yes!

Figure 2

caption. Would be helpful to say which scaling scheme used (Lm, St, LSD?).

We have included both the scaling scheme (LSD) and calibration data (Promontory Point) used to calculate $^{10}$Be exposure ages.

Also, in text, you say there are bedrock samples? Which are bedrock? Caption states all red circles are boulders? Maybe use a different color symbol?

Yes, we see how this is unclear. We have, per your suggestion, coded the bedrock samples from Pine Creek by color and have updated the caption text as follows: "Moraine deposits are shown in yellow, cosmogenic 10Be boulder and bedrock sample locations are indicated by red and green circles, respectively, with exposure ages (Promontory Point calibration data and LSD scaling scheme) and analytical uncertainty (shown in ka) and sample codes in blue text."

Also, relevant to a comment below, can you say if there are Bull Lake moraines shown on Figure 2? If so, how far from the Pinedale as mapped? I cannot help but ask (see below), given the info

provided in this manuscript, whether any Bull Lake deposits if they exist (?) could be early Pinedale?

For example, in top image are there right lateral Bull Lake moraines visible in imagery?

If yes, I know too late, but one or two samples in future work would help determine if indeed they are Bull Lake or early Pinedale? If Bull Lake does not exist in field area, obviously my comment is not relevant.

Yes, it would be fantastic to have found and dated Bull Lake deposits in the area. Unfortunately, none were mapped in the study area.

Lines 203+     Maybe I am missing it, but can you mention where your moraine cosmo samples are in the context of frontal or lateral moraines? If only in the frontal moraines, for example, could there be less earlier landforms preserved (due to later outwash, etc)? That is, versus the laterals? Even a sentence or two.

We see how this may be ambiguous and have updated the text  in the Methods section for clarity as follows: "Following moraine mapping and field verification, we selected moraines and erratic boulders atop moraine crests for in-situ cosmogenic 10Be exposure dating to determine landform ages. at Cut Bank Creek, South Fork Deep Creek, and Cascade Creek canyons. We targeted two frontal moraines at Cut Bank Creek including the ice-distal terminal moraine. Boulders atop a recessional moraine identified just beyond the mouth of Cut Bank Canyon were also sampled to limit the time when moraine building at the mountain front ceased and ice retreat commenced. At South Fork Deep Creek and Cascade Creek, we targeted lateral moraines associated with maximum mapped ice extents. We collected samples from bedrock and erratic boulders not associated with moraine deposits along a transect of Pine Creek canyon"

206. typo/grammar

238 – was this measured more precisely?

No.

In the Cosmo section

A little more info is needed here or in background. Semi arid? So, relatively dry so not a ton of snow to cover boulders through spring? Are boulders in a forest? Or areas sampled not forested (one image has a thin forest). Any samples above tree line, so not have to worry about trees or forest floor (e.g.,snow, fire)

I think I understand what the reviewer is getting at and have added the following text to the Methods section: "We do not account for snow cover shielding on calculated exposure ages, noting that snow shielding corrections (~3%; Marcott et al., 2019) are typically less than the total uncertainty associated with production rate calibration and scaling schemes (~6%; Balco, 2020)."

More important though: 1) While I agree in principle with the idea of using a local production rate calibration (promontory point), I recall this rate was slightly higher (or at least different) than other rates. While this may be correct, and it is indeed higher here (compared to the rest of the world?), I think the authors should at least use another production rate for comparison and mention this. Maybe the SPICE rates (Fenton et al) or even one of the other coherent rates at mid-high Northern Hemisphere latitudes.

This is a fair point. We have included exposure ages calculated with Promontory Point and Primary (e.g. Borchers et al., 2016) calibration data and LSD, St, and LM scaling in an additional supplemental table.

2)      the authors need to give St, Lm, and LSD in a table. All three scaling schemes. There is a reason the output gives all three scaling schemes, especially still Lm/St.

3)      , and say what is the difference between them? Between LSD and Lm, for example? So reader knows what is possible range?

Now, it is possible LSD is the most accurate for this part of the globe, and some would say likely, however, if you can make a statement that differences are only a few (?) or 5%, that would help the reader see it does not even matter.

Yes agreed. We again have added a supplemental table that addresses these concerns and added the following analysis to the Methods section text: "Exposure ages calculated using other commonly used calibration (e.g. CRONUS-Earth primary 10Be calibration data; Borchers et al., 2016) and scaling schemes (Stone et al., 2000) result in apparent age differences of 3-5%."

284 to 296. Can you give a reader an idea how much this 'secondary stuff' really matters?

<5% difference in your final values (assuming reasonable variability of these tuning knobs)?

Yes, we have updated the discussion of model uncertainty to indicate that the uncertainties used were derived from sensitivity analyses – thus indicating how much the 'secondary stuff' matters.

374 – which on figure 2 are bedrock samples? Is everything not on yellow in figure 2 bedrock? Perhaps use a different symbol color on figure 2?

Yes we have updated figure two to reflect bedrock and boulder sampling locations.

Figure 4. What are the errors? The internal or external uncertainties in the table? For comparison to each other from the same field areas, I would only show the internal errors on this plot, which is really just showing us the results compared to one another. Also, I have no sense of whether Lm or St are significantly different, if you are to use those schemes – even if you have a reason to prefer LSD (fair enough).

The internal uncertainty is the analytical uncertainty. In the figure caption, we state that exposure ages are plotted with analytical uncertainties.

Yep, bedrock notoriously has inheritance, even at middle latitudes. Not surprised.

Table 1. I would have a separate Table 2 with all three scaling schemes, St, Lm, and LSD. There is a reason they are output. If you feel the need for space issues, perhaps put in supplement with AMS ratios.

We have done exactly this.

Figure 5. the modeling adds a very nice touch to the study.

463. Just to confirm double check, you said earlier you recalculated literature ages (licciardi and Pierce) and are also presenting these with promontory point production rate, in LSD space?

Every exposure age reported uses PP calibration and LSD scaling except in the newly added supplemental.

More important though, 17.5 +/- 0.6 and 18.2 +/-0.5 overlap at 1 sigma (or 1 SEM), so you cannot say they are different. I agree, taken at face value it seems so, but statistically you cannot. The 16.9 ka is younger indeed.

Good point. We have updated the text as follows to reflect this clarification: "The $^{10}Be$ exposure ages presented here for the South Fork Deep Creek (17.5 ± 0.6 ka) and Cascade Creek (16.9 ± 0.1 ka) lateral moraines in the northern Absaroka Range appear slightly younger than ages from the previously dated lateral moraine in the neighboring Pine Creek valley in the

northern Absaroka (10Be exposure age = 18.2 ± 0.5 ka, with the standard error of ages recalculated from Licciardi and Pierce, 2008) but do overlap within one standard error."

I would give an (n=x) after each 'ka' when presenting the mean ages. So, the reader has an idea of the robustness of the mean.

Yes, we agree it is important for the reader to have an idea of how robust the landform/abandonment ages are. We have already done this in the Results – Cosmogenic $^{10}$Be Exposure Ages section (lines 426-438).

Also, you could just compare the analytical (internal) uncertainties, as the sites are so close to each other the differences in scaling probably can be considered negligible.

Moraine abandonment/landform age uncertainties are reported as the standard error of the mean.

471 moraines…?

Yes.

479. See comments above about this comparison. What is global as defined by Clark et al in relation to outer middle pinedale?

We indicate the global LGM age range as 26.5-19.0 ka and can reiterate here.

536-537. Could middle Pinedale just be the same extent as early Pinedale, so you do not see the latter preserved?

This is a good point and one that we only implicitly made. We have updated the text to state this explicitly as follows: "This pattern is observed throughout the Rocky Mountains and suggests that the mountain glacier moraine chronology in western Montana differs from the rest of the region, such that the outermost moraines do not represent the early Pinedale interval and only represent the middle Pinedale interval suggesting similar or more extensive ice during the middle as compared to the early Pinedale. This may reflect the importance of regional climatic effects on mountain glaciation, especially the strengthening of westerly airflow and attendant moisture delivery, as described above."

Also, I do not have a sense of where the 'inferred' Bull Lake is. It is not shown on Figure 2 or 1. Can you rule out that some of what has been traditionally mapped as Bull Lake in this area could

be early Pinedale? Not that I am arguing for this, and the authors may see it as an extreme devil's advocate position, but as I do not have any sense of where the Bull Lake is on Figure 2, so I ask the question.

It is always possible that deposits previously mapped as Bull Lake are, in fact, early Pinedale deposits. However, we did not map or collect samples from any Bull Lake deposits.

566. Mumma et al (2012) attributed.

Updated

Figure 6. The modeling work is very nice and authors should be commended. One question – is the modeling giving an 'annual temperature' ? On plots such as this, and in the text, is the reader seeing annual temperature or do authors interpret this as an annual temperature? Or a summer temperature (which drives ablation)? This might make the discussion even more informative than it is already.

This is an important point and one we thank the reviewer for bringing up. First, we have made this explicit in the Method section Line 293: "It is important to note that in our simulations we change monthly temperature and precipitation distributions for the entire year while glacier mass-balance is primarily sensitive to ablation season temperatures and accumulation season precipitation".

608-616. See comment above for Figure 6 regarding how the reader should think about these temperatures – annual or summer? I assume PMIP is annual.

647. …range…

Updated.

….648. Stadial…

Figure 7. Just remind reader that squares and error bars represent 10Be ages +/- 1s (or something else?). they mention the 18.2 ka, so I have impression we are looking at both mean ages and individual bedrock ages plotted? Hence, just please clarify.

Good point. We have updated the Figure 7 (now Figure 8) extensively, including the use of a age-depth model with Monte Carlo uncertainty analysis, with new caption: "Figure 8. Age-distance (top) and age-elevation (bottom) ice retreat models for Pine Creek. Bedrock exposure ages plotted with analytical uncertainties while the Pine Creek moraine age uncertainty is shown as the standard error. In both transects, the data are relative to the Pine Creek headwall. Dark curves represent median age models while dashed lines indicate boundaries of 95% confidence interval (CI)."

557. need to define lateglacial somewhere. I get the impression authors are discussing more than just YD and ACR time? Not sure, hence please clarify.

Yes this is ambiguous as written. We have updated the text at the first use of the term 'Lateglacial' (Line 682) as follows: "However, the coherence of ice-retreat rates in the Absaroka Range with locations across the Rockies from ca. 16 ka through the Lateglacial (i.e. (19–11.7 ka; Reitner et al., 2016) suggests common factors driving deglaciation across the region. For example, glacier retreat in Rocky Mountains after ca. 16 ka coincides with sustained increases in atmospheric CO2 and regional temperature changes despite some glacier retreat lagging behind initial rises in CO2 around 17 ka (Figure 8)."

662. timing of terminal moraine abandonment. Please clarify precisely what is meant. Do authors mean findings of early Pinedale elsewhere versus what they found? Or

We have updated the text as follows: "While the timing of initial abandonment of ice-distal positions is variable at sites across the Rockies, which range from the end of the global LGM to ca. 16 ka, the broad pattern and timing of subsequent deglaciation after ca. 16 ka is similar across the Rocky Mountains (Figure 8)"

abandonment of terminal moraines no matter what their age in a given area, terminal moraine abandonment whether 23ka or 18-17 ka?

Or do they mean – as I understand - instead that the timing of terminal or maximum moraine formation is variable….? The timing of maximum glacial expansion? Hence, writing not clear.

677 – same comment as prior.

678. Somewhere define lateglacial as they use it.

Yes, thank you for catching this. We define Lateglacial at first use (Line 709) as 19–11.7 ka; Reitner et al., 2016.

Figure 8. I wonder if panel D would be more readable or visually better to have a legend on the left side (instead of defining all these things in the caption). Just a suggestion.

Other

Define what normalized against? Normalized (or standardized) to what?

We have updated the figure caption text as follows: "…(D) Normalized glacier elevation (i.e. 1 = terminus, 0 = headwall) for Pine Creek glacier in the Absaroka Range (red stars), Teton Range, WY (green boxes), Wasatch Range, UT (black diamonds), Uinta Mountains, UT (white triangles), Front Range, CO (blue inverted triangles), Sawatch Range, CO (blue circles), and San Juan Range, CO (blue squares) based on cosmogenic exposure dating…"

Also, remind reader if panel D items are based on cosmo dating or modeling? Discussed in text, but maybe good to remind reader briefly how these symbols are plotted (model results? Cosmo based?)

693. See comments above about LGM. I am not sure what this means – is this then a local LGM?

I think much of the confusion regarding our use of the term LGM can be alleviated by replacing with the phrase 'global LGM', in other words the LGM as defined by Clark et al. We have updated throughout the text accordingly.

704 – see above comment about temperature. 707 typo.

Supplement. Need legend of colors and other info – black line, white line, etc. Also, probably should add what background image is. May be obvious, but cannot hurt to repeat it, so figure cation stands alone, reader does not have to go back through main paper to understand figure.

Yes, we have updated the figure caption text as follows: "Supplemental Figure 1. Model results for Cut Bank (-9.2°C) and Lake Creek (-8.0°C) glaciers at maximum Pinedale extents with 100% modern precipitation. Black lines indicate mapped glacier extents the simulations attempted to match."

Supplement table.

1) If you have the concentration or density of the spike added (carrier), that would be almost equally important (not quite, but useful), to add. This would be in ppm typically. In text you just give a value of 1000 ppm I recall – was it measured more accurately?

We have restated in the supplemental table the concentration of the spike used. We used the certified concentration and did not measure independently.

Also, please add whether you subtracted 10/9Be ratios of blanks from 10/9Be ratios of samples, or you calculated 10Be atoms actually in the blank, and subtracted this # from 10Be atoms in each sample?

We have added the following line to the Results section (Line 387): "We corrected sample $^{10}$Be/$^{9}$Be ratios by subtracting the number of $^{10}$Be atoms in the corresponding blank from the sample."

---

## Author Comment (AC2)

Clim. Past Discuss., referee comment RC2
https://doi.org/10.5194/cp-2021-106-RC2, 2021

[Figure]

**Comment on cp-2021-106**

Anonymous Referee #2
* * *
Referee comment on "Late Pleistocene glacial chronologies and paleoclimate in thenorthern Rocky Mountains" by Brendon Quirk et al., Clim. Past Discuss., https://doi.org/10.5194/cp-2021-106-RC2, 2021
* * *
This is an interesting study, that combines glacial geology and glacial modelling to infer past glacial conditions in the Northwestern USA. This combined approach (not common in the literature) is always welcome and can provide significant insights regarding the climate evolution of the planet. The scope and structure of the paper is sound, and the outcome could eventually be of major interest for the scientific community. However, the manuscript, in its present form, needs some revision in order to improve some aspects. I would like to invite the authors to consider the comments below, which I hope will help improve the manuscript. Finally, I would like to mention that this manuscript would benefit significantly from a review by a glacial modeler.

General comments:

I am not a native English speaker; however, I can recognize the need for some tidying up of the wording of some sentences throughout the manuscript.

When introducing new areas or sites, please provide coordinates (at least latitudes). People outside the US are not necessarily familiar with the locations discussed in the text.

> We have done this.

I am a little concerned with the way that the authors treated outliers (see specifics below). I don´t know if this has a major impact on the main conclusions, but it needs to be addressed consistently.

Agreed. See comments below.

I don´t agree with the discussion in the section "The pace of ice retreat in the Rocky Mountain". If you put all the ages together (e.g., figure 7), it is clear that the ages from upstream and downstream are statistically indistinguishable (even at 1 sigma). As such, all the associated analysis falls apart. The good news is that this manuscript (an all the main conclusions) do not depend on this section. I suggest removing this section.

I believe this interpretation does not fully consider the law of superposition. In other words, we have additional context besides the ages themselves, such as distance, elevation and morphostratigraphy, that provide relative age constraints. However, the comment does raise an important point regarding uncertainty analysis. We have performed additional analyses to quantify this uncertainty (see figure below). We adapt the COPRA algorithm (Breitenbach et al., 2012) to quantify the uncertainty in our reconstructed age- distance and elevation transects. Here, we have estimated uncertainty using 10,000 Monte Carlo realizations satisfying superposition.

[Figure]

Comments line by line:

INTRODUCTION:

56-58: why is this important? Maybe add 1-2 sentences

We have modified the text as follows: "These spatiotemporally constrained paleo-glaciers can then, in turn, be used to infer paleoclimate conditions in the northern Rocky Mountains during the last glaciation **for which relatively few records exist compared to other regions of western North America**"

84-85 Awkward wording

We have modified the text as follows: "The Lewis Range (48.5°N, 113.5°W) hosted numerous glaciers during the latest Pleistocene and, in some areas, these glaciers coalesced to form the northern Rocky Mountain ice cap (Locke, 1995; Figure 1)".

89-91: this should be in the result section (justified by evidence) or cited from previous publications.

We have cited the relevant publication.

96: what do you mean that they generally flowed down to elevations of 1.6 km? When? 98-101: are we talking during the last glacial cycle?

We have updated the text as follows: "In this study, we focus on the Cut Bank Creek glacier which flowed east from its headwaters at 2.6 km asl and terminated on the piedmont just above 1.4 km asl at its maximum extent. The Cut Bank glacier did not coalesce with either the northern Rocky Mountain ice cap to the west and north or the Laurentide ice sheet to the east during Pinedale times and flowed as a discrete mountain glacier (Calhoun, 1906; Alden, 1932)"

92-101: I am missing citations…. How do you know all this information? (e.g., ice thickness). I couldn´t find a single reference in the Site Description section. Has anyone else worked in the area?

We have updated the section to include the appropriate references.

104: "and relatively little work has been done inferring past climate in the region from paleoglacier characteristics". You should include some examples (cites) at the end of the sentence.

Yes, agreed. In the text as written, the following paragraph cites and describes the published work that has been done on this topic.

128: Did you recalibrate these ages with the latest curves? 133-135: are these phases progressively less extensive?

Yes. We have updated the text as follows: "No numerical ages are available for these deposits, although a radiocarbon age on a wood fragment, underlying two latest Pleistocene tephra layers in lake sediment at Marias Pass, provides a minimum age of 12,194±145 14C yr (Carrara, 1995) or 13.8-14.8 cal kyr (Fullerton et al., 2004; recalibrated here using IntCal13 (1σ); Reimer et al., 2013) for complete recession of at least one east-side outlet glacier of the Northern Montana Ice Cap in the Glacier National Park region."

138-141: very awkward wording. Please, rephrase.

Agreed. The text has modified as follows: "Terminal and recessional moraines at the southwestern front of the northern Absaroka Range and in Paradise Valley to the south have cosmogenic 10Be exposure ages that were originally reported by Licciardi et al. (2001) and have been supplemented with additional data from Licciardi and Pierce (2008; 2018)"

146-149: you need to discuss the meaning of the 10Be ages (how do you interpret them?) before presenting this statement. Are they minimum ages of stabilization? Close-minimum ages for the retreat? Maximum ages?

We have made this explicit in the text and have modified as follows: "Here, we interpret exposure ages as ice retreat or moraine abandonment ages. Thus, the exposure ages from the Greater Yellowstone glacial system suggest that mountain glaciers began retreating from their terminal moraines during the middle Pinedale and after the end of the global Last Glacial Maximum."

152 define late Pleistocene

This is a good point considering the Late Pleistocene often refers to time periods much older than those considered here. We have updated the text as follows: "While many investigations in western Montana have focused on reconstructing the extent and chronology of the Pinedale glaciation, fewer have attempted to describe Pinedale climate conditions." Please note that the Pinedale glaciation is defined at Lines 121-124

164-170: the wording makes it difficult to understand the point of this long sentence.

Agreed, we have updated the text as follows for clarity: "Modern methods used to reconstruct paleo-glaciers, particularly distributed energy/mass-balance or degree-day mass-balance models,

have been successfully applied to sites in the Middle (Laabs et al., 2006; Refsnider et al., 2008; Birkel et al., 2012; Quirk et al., 2018, 2020) and Southern Rocky Mountains (Ward et al., 2009; Brugger, 2010; Brugger et al. 2018, 2019; Dühnforth and Anderson, 2011; Leonard et al., 2014, 2017a; Schweinsberg et al., 2016). In this study we apply a modified version of the Plummer and Phillips (2003) distributed energy/mass-balance model to reconstructed glaciers in the Absaroka and Lewis ranges to help elucidate climate conditions in the northern Rockies during the last glaciation."

METHODS

181-183: move to results

Figure 2: for clarity, please choose a different color for the outline of recessional positions 206: delete "."

255: why did you choose standard error of the mean instead of standard deviation?

It's a better representation of landform age uncertainty as opposed to individual boulder scatter.

269-271: To do that, you need to assume that the moraines in both valleys are coeval, correct? If so, you should mention it in the text. Or am I missing something?

This is discussed in the results section as possible source of error (i.e. temporal offset).

339: A detailed description of the geomorphology of the area is presented; however, it is very difficult to visualize /assess it, given that no detailed geomorphological map is presented (except figure 2 where the authors only depict the moraines)

Yes, we agree with the reviewer's comment. We have included a detailed geomorphological map of the Cut Bank Creek area as the new Figure 4.

356-359: it is unclear to me what the authors want to say here… outer and inner moraines?

We have updated the text for clarity as follows: "The moraines delimit the size and shape of the piedmont lobes formed by glaciers in the two valleys. At Cut Bank Creek, the piedmont lobe had a maximum diameter of 6.8 km while occupying the outer, ice-distal moraine. While occupying the ice-proximal sector, delimited by the mapped recessional moraine, the piedmont lobe was reduced in diameter to approximately 4.4 km and likely became thinner or formed a more gradual slope near the terminus as evidenced by the lower relief along the moraine"

355-367: why do you focus your description only on glacier width? What about glacier extent? Area?

We agree with the reviewer and it was an oversight to not include descriptions of glacier areas and lengths. We have therefore modified the text as follow to include these important metrics: "The moraines delimit the size and shape of the piedmont lobes formed by glaciers in the two valleys. At Cut Bank Creek, the maximum Pinedale glacier, as denoted by the ice-distal moraine, extended almost 30 km from the headwall and occupied an area of ~123 km2 while the piedmont lobe had a maximum diameter of 6.8 km. While occupying the ice-proximal sector, delimited by the mapped recessional moraine, the Cut Bank glacier extended approximately 25 km down valley and occupied an area of 86 km2 while the piedmont lobe was reduced in diameter to approximately 4.4 km and likely became thinner or formed a more gradual slope near the terminus as evidenced by the lower relief along the moraine. The piedmont glacier width was further diminished upon retreat to the recessional moraine to approximately 1.3 km, only slightly wider than the mouth of Cut Bank Canyon . In Lake Creek valley, the piedmont lobe formed an irregular shape, likely due to partial confinement of the northern side of the lobe by the right-lateral moraine in the neighboring Cut Bank Creek valley. The piedmont lobe had a maximum width of about 2.5 km, a total glacier length of 12 km, and occupied an area of 24 km2 when the terminal moraine was occupied. Upvalley of the terminal moraines in Cut Bank Creek and Lake Creek valleys, lateral moraines and other glacial features mapped by Carrara (1989) were used to delimit ice thickness and areal extent."

Fig 4: it would be ideal to see the rest of the samples in this plot as well

We understand the reviewer's request. However, and in consideration of the reviewer's other salient comments, we do not think it is appropriate to include the ice-retreat exposure ages on a plot that does not show the spatial relationship between samples. This has, however, been done in Figure 8.

Fig. 4: I don´t see the logic behind considering CB12 as an outlier and not doing the same with sample DC1204. Neither of those overlap at 1 sigma. On the other hand, if you use 2 std, none of these samples would be considered outliers.

Yes, we agree, there is a lack of consistency in how we determined outliers. Thus we treat DC1204 as an outlier.

425: is that even possible? 3x precipitation? Hard to assess since the authors didn´t provide present day values in the site description… 3x the precipitation would be equivalent to 1std? 2std? 10 std? Maybe, such analysis could help to refine the results.

This is a good point. We have included a brief description of modern climatology to the 'Site Description' section. More broadly, the modeled precipitation changes are selected in order to define a curve in precipitation-temperature (P-T) space that represents a given glacier stadial. Thus, the individual points need not necessarily be solutions. Further, the P-T curve for a glacier can be modeled with an exponential and it is therefore advantageous to select points spaced reasonably far apart to capture the exponential behavior.

Fig 5: (This comment may be out of ignorance since I am not a modeler) I understand that it is almost impossible to exactly match the modelling results to field evidence, however in panel A I see plenty of room for a bigger drop in temperature, at both the glacier front and headwalls. Can you explain?

Small increases (i.e. tenths of a degree) in temperature depression or precipitation % result in mismatch between simulated and mapped extent – particularly along the piedmont boundary. This is, unfortunately, the best we were able to simulate the paleo-glacier shape.

DISCUSSION

461-464: Actually, 17.5±0.6 ka and 18.2±0.5 ka are statistically indistinguishable. Furthermore, you never discuss if SF Deep Creek and Cascade Creek are coeval or not.

Following corrections for how we determine outliers, text should now address this more clearly: "The 10Be exposure ages presented here for the South Fork Deep Creek (18.1 ± 0.1 ka) lateral moraine agree well with the landform age from the previously dated lateral moraine in the neighboring Pine Creek valley in the northern Absaroka (10Be exposure age = 18.2 ± 0.5 ka, with the standard error of ages recalculated from Licciardi and Pierce, 2008). The Cascade Creek (16.9 ± 0.1 ka) moraine exhibits a younger age than both the South Fork Deep Creek and Pine Creek moraines. However, and as previously discussed, the Cascade Creek moraine exposure ages should be considered preliminary due to relatively high analytical uncertainties. Although these moraines were deposited by discrete valley glaciers, their exposure ages are similar to 10Be exposure age of the nearby Eightmile terminal moraine (17.9 ± 0.4 ka, recalculated from Licciardi and Pierce, 2008), the outermost moraine of the last glaciation deposited by the northern outlet glacier of the Yellowstone Icecap, as well as to the age of the Chico moraine (17.1 ± 0.6 ka recalculated from Licciardi and Pierce, 2008) the initial moraine deposited during recession of this outlet glacier. These ages for the outermost and initial recessional moraines in the northern Yellowstone/northern Absaroka Range area in southwestern Montana are also very similar to those we report here for the terminal (17.2 ± 0.2 ka) and initial recessional (16.4 ± 0.2 ka) moraines at Cut Bank Creek in northwestern Montana. Taken together, these ages suggest that terminal moraines in western Montana were occupied until ca. 18-17 ka and that glaciers

were still near their maximum lengths at ca. 17-16 ka in northern Yellowstone and in the Lewis Range, as indicated by exposure ages of the recessional moraines."

541: of the for the…. Review this sentence 668-670: citations for $CO_2$ increase?

We have corrected the typo, thank you for pointing it out. We have included references to the $CO_2$ data in the text and figure caption.